# IMAGENHUB: STANDARDIZING THE EVALUATION OF CONDITIONAL IMAGE GENERATION MODELS

**Max Ku♠, Tianle Li♠, Kai Zhang†, Yujie Lu♣, Xingyu Fu♡, Wenwen Zhuang◇, Wenhu Chen♠**
University of Waterloo♠, Ohio State University†, University of California, Santa Barbara♣
University of Pensylvania♡, Central South University◇
{max.ku, t29li, wenhuchen}@uwaterloo.ca

https://tiger-ai-lab.github.io/ImagenHub/

## ABSTRACT

Recently, a myriad of conditional image generation and editing models have been developed to serve different downstream tasks, including text-to-image generation, text-guided image editing, subject-driven image generation, control-guided image generation, etc. However, we observe huge inconsistencies in experimental conditions: datasets, inference, and evaluation metrics – render fair comparisons difficult. This paper proposes ImagenHub, which is a one-stop library to standardize the inference and evaluation of all the conditional image generation models. Firstly, we define seven prominent tasks and curate high-quality evaluation datasets for them. Secondly, we built a unified inference pipeline to ensure fair comparison. Thirdly, we design two human evaluation scores, i.e. Semantic Consistency and Perceptual Quality, along with comprehensive guidelines to evaluate generated images. We train expert raters to evaluate the model outputs based on the proposed metrics. Our human evaluation achieves a high inter-worker agreement of Krippendorff's alpha on 76% models with a value higher than 0.4. We comprehensively evaluated a total of around 30 models and observed three key takeaways: (1) the existing models' performance is generally unsatisfying except for Text-guided Image Generation and Subject-driven Image Generation, with 74% models achieving an overall score lower than 0.5. (2) we examined the claims from published papers and found 83% of them hold with a few exceptions. (3) None of the existing automatic metrics has a Spearman's correlation higher than 0.2 except subject-driven image generation. Moving forward, we will continue our efforts to evaluate newly published models and update our leaderboard to keep track of the progress in conditional image generation.

## 1 INTRODUCTION

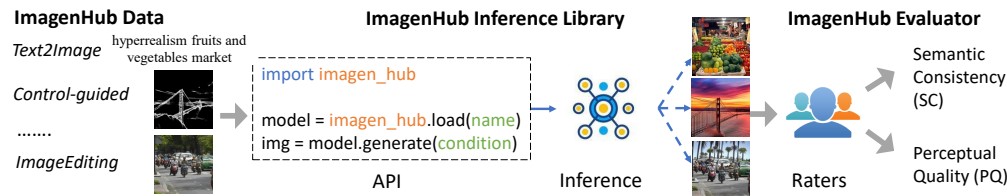

Figure 1: The overview of ImagenHub framework, which consists of the newly curated ImagenHub dataset, ImagenHub library, and ImagenHub evaluation platform and standard.

With the recent development of diffusion models, image generation has quickly become one of the most popular research areas in AI. To enable controllability in the image generation process, a myriad of conditional image generation models have been proposed in the past year. Diverse set of conditions have been attempted to steer the diffusion process. One of the most popular tasks is text-guided image generation (Ramesh et al., 2022; Rombach et al., 2022; Saharia et al., 2022),

which aims to ground on a text prompt to generate the corresponding image. Besides that, there are also subject-conditioned image generation (Gal et al., 2022; Ruiz et al., 2023), text-guided image editing (Brooks et al., 2023), multi-subject-conditioned image generation (Kumari et al., 2023), style-guided image generation (Sohn et al., 2023), etc. These different tasks aim to serve different types of downstream applications by enabling subject-level, background-level, style-level controls. The field is evolving at an unprecedented pace and lots of improvements have been reported in the published papers. However, one glaring issue we observed is the published work are highly inconsistent in their experimental setups. To summarize, the inconsistencies mainly come from three aspects, namely dataset, inference and evaluation:

- **Dataset**: The existing work curated their own evaluation dataset, which makes the comparison of different models totally incomparable.
- **Inference**: Some work requires heavy hyper-parameter tuning and prompt engineering to achieve reasonable performance, which makes the model less robust. Due the tuning effort on different models differ significantly, their comaprison could become unfair.
- **Evaluation**: The existing work used different human evaluation protocols and guidelines. This inconsistency renders it impossible to compare human evaluation scores across different methods. Moreover, some of the work either employs a single rater or does not report inter-worker agreement. Thus, the reported results might not be comparable across different papers.

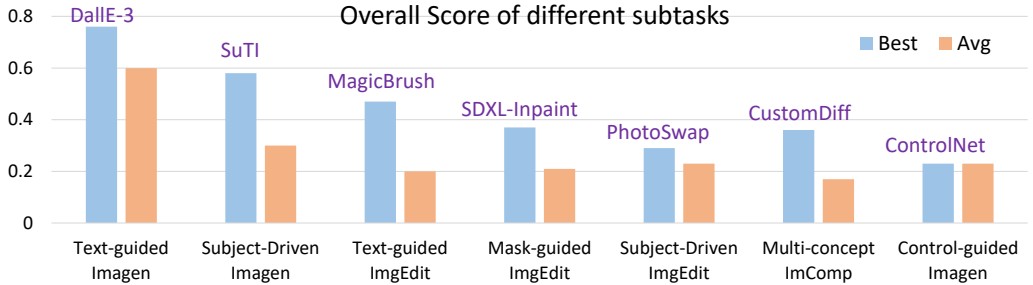

Figure 2: The best and the average model performance in each task

These three inconsistencies make it nearly impossible to track the real progress in the field of the conditional image generation. Such an issue could greatly hinder the development of this field. The desiderata is to build a centralized effort to fairly evaluate every model. More importantly, this effort needs to be continuous to keep up with the evolvement of the field. Our paper aims to serve this purpose to standardize the serving and evaluation of all open-source conditional image generation models. More specifically, `ImagenHub` consists of the modules listed in Figure 1.

`ImagenHub` **Dataset**. We surveyed the existing public evaluation sets for all the generation tasks and then picked diverse instances from them to build our `ImagenHub` dataset. `ImagenHub` dataset consists of 7 subsets, each with 100-200 instances. This dataset aims at standardizing the evaluation input to ensure fair comparison for different models.

`ImagenHub` **Inference Library**. We built a `ImagenHub` library[1] to evaluate all the conditional image generation models. We ported the highly dispersed codebase from the existing works and then standardized them into a unified format. During inference, we fixed the hyper-parameters and the prompt format to prevent per-instance prompt or hyper-parameter tuning, which makes the inference of different models fair and reproducible. The library is designed to be easily extendable. Our Appendix A.7, A.8, and A.9 show how a third party and researchers can benefit from our work.

`ImagenHub` **Evaluator**. We explored different human evaluation metrics and iterated over different versions of the rating guidelines to improve the inter-rater agreement. We settled on two three-valued rating metrics 'Semantic Consistency' and 'Perceptual Quality' to achieve generally high inter-worker agreement measured by Fleiss Kappa (Fleiss & Cohen, 1973) and Krippendorff's Alpha (Krippendorff, 2011). We designed a rating standard to achieve several benefits: (1) Our rating guide is an unambiguous checklist table such that the rater can rate the image with ease. (2) The

---

[1]it's similar to Huggingface libraries (Wolf et al., 2019; von Platen et al., 2022)

designed rating guideline is unified on every task type. (3) Sustainability. Since each model is rated individually, previous evaluation results can be reused when new models are added.

We demonstrate our evaluation results in Figure 2, where we show the overall score of the best-performing model and the medium-performing model. Based on our evaluation results in section 5, we made some general observations:

- The existing models' performance is generally unsatisfying except for Text-guided Image Generation and Subject-driven Image Generation.
- We found that evaluation results from the published papers are generally consistent with our evaluation. 83% of the published result ranking is consistent with our ranking.
- Automatic evaluation metrics do not correlate well with human preference well except subject-driven image generation. The correlation scores are lower than 0.2.

## 2 THE PROBLEM OF CONDITIONAL IMAGE GENERATION

The goal of conditional image generation is to predict an RGB image $y \in \mathcal{R}^{3 \times H \times W} : \mathcal{Y}$, where $H$ and $W$ are the height and width of the image. The prediction of the target image is given a set of input conditions $X = [c_1, c_2, \cdots]$, where $X \in \mathcal{X}$, where $C_i$ denotes the i-th condition. In the problem, we aim at learning a prediction function $f : \mathcal{X} \to \mathcal{Y}$ with deep learning models. Here we mainly consider $f$ parameterized with diffusion models. Here we list a set of tasks we consider in Figure 3, where $c_i$ can be represented as text prompt, image mask, subject image, source image, background image, control signal, etc.

**Task Definition.** We formally define the tasks we consider as follows:

- Text-guided Image Generation: $y = f(p)$, where $p$ is a text prompt describing a scene. The goal is to generate an image consistent with the text description.
- Mask-guided Image Editing: $y = f(p, I_{\text{mask}}, I_{\text{src}})$, where $I_{\text{mask}}$ is a binarized masked image, and $I_{\text{src}}$ is a source image. The goal is to modify given $I_{\text{src}}$ in the masked region according to $p$.
- Text-guided Image Editing: $y = f(p, I_{\text{src}})$. It's similar to Mask-guided Image Editing except that there is no mask being provided, the model needs to figure out the region automatically.
- Subject-driven Image Generation: $y = f(S, p)$, where $S$ is a set of images regarding a specific subject, which normally ranges from 3-5. The goal is to generate an image according to $p$ regarding the subject $S$.
- Subject-driven Image Editing: $y = f(S, p, I_{\text{src}})$, where $I_{\text{src}}$ is a source image and $S$ is the subject reference. The goal is to replace the subject in $I_{\text{src}}$ with the given subject $S$.
- Multi-concept Image Composition: $y = f(S_1, S_2, p, I_{\text{src}})$, where $S_1$ and $S_2$ are two sets of concept images. The goal is to compose them together to generate a new image according to the text description $p$.
- Control-guided Image Generation: $y = f(I_{\text{control}}, p)$, where $I_{\text{control}}$ is the control signal like depth map, canny edge, bounding box, etc. The goal is to generate an image following the low-level visual cues.

## 3 RELATED WORK

**Multimodal Conditional Image Synthesis.** Recent works in multimodal conditional image synthesis often rely on pre-trained vision-language models such as CLIP (Radford et al., 2021) and pre-trained large diffusion models like Stable Diffusion (Rombach et al., 2022). CLIP bridges the gap between textual descriptions and visual content, while Stable Diffusion is a latent diffusion model trained on a massive dataset. CLIP and Stable Diffusion are now widely used as core components in applications such as text-to-image generation, text-guided image editing (Patashnik et al., 2021; Meng et al., 2021; Couairon et al., 2022; Mokady et al., 2022; Zhang et al., 2023), and mask-guided image editing (Avrahami et al., 2022; Nichol et al., 2022). Another interesting research direction is image subject personalization with a few or even one image of the subject. Textual Inversion (Gal et al., 2022) optimizes a text token vector to represent the subject, while DreamBooth (Ruiz et al., 2023) finetunes the Stable Diffusion model to learn the new subject concept efficiently. These works foster research in potential applications like subject-driven image generation and editing (Li et al., 2023a;b; Gu et al., 2023), and even multi-concept image compositions (Kumari et al., 2023). To explore more control conditions, ControlNet (Zhang & Agrawala, 2023) proposed the usage of zero

| $c_1$ | $c_2$ | $c_2$ | Task | $y$ |
|---|---|---|---|---|
| A cartoon styled alarm clock | $\emptyset$ | $\emptyset$ | Text-to-Image Generation |  |
|  |  | Change frisbee to a football | Mask-guided Image Editing |  |
|  | Make it a slice of pizza instead of the sandwich | $\emptyset$ | Text-guided Image Editing |  |
|  | A [V] dog in the Versailles hall of mirrors | $\emptyset$ | Subject-Driven Image Generation |  |
|  |  | Replace glasses with [V] glasses | Subject-Driven Image Editing |  |
|  |  | A cat [V] standing by a pot [M] | Multi-Concept Image Composition |  |
|  | A small dog is curled up on top of the shoes | $\emptyset$ | Control-guided Image Generation |  |

Figure 3: The visualization of all the conditional image generation tasks. Here we consider tasks with 1-3 conditions, where $\emptyset$ means empty. The special token [V] and [M] are special identifiers.

convolution on the Stable Diffusion model to support additional guided image control. This work brings up the idea of the control-guided image generation task and inspired later work (Qin et al., 2023) on improving the control versatility.

**AI-generated Image Assessment.** Evaluating AI-generated images holistically is a complex and open problem (Salimans et al., 2016). Researchers have proposed various automatic metrics. In the image quality aspect, Inception score (Salimans et al., 2016), FID (Heusel et al., 2017) are often used. These methods rely on statistics from an InceptionNet pre-trained on the ImageNet dataset. Despite being widely adopted due to their sensitivity to small changes in images, these metrics are not ideal. They are biased towards the ImageNet dataset, resulting in inadequate evaluations (Borji, 2021). Later works like LPIPS (Zhang et al., 2018) and DreamSim (Fu et al., 2023) proposed better ways to measure the perceptual similarity. In the semantic consistency aspect, the CLIP score (Hessel et al., 2021) is often used to measure the vision-language alignment between the generated image and the prompt. Researchers also worked on alternative methods such as BLIP score (Li et al., 2022) and ImageReward (Xu et al., 2023). However, in some tasks like subject-driven image generation and editing, the automatic measurement of semantic consistency is still an open problem. One long-established yet effective approach to assessing AI-generated image performance is to rely on human annotators to assess the visual quality (Denton et al., 2015; Isola et al., 2017; Meng et al., 2021; Chen et al., 2023). The downside is that it entails a reliance on human judgment, which can introduce subjectivity and potentially limit scalability. To mitigate the downsides, the human evaluation design has to be unambiguous and easy to follow.

# 4 METHOD

## 4.1 HUMAN EVALUATION METRICS

Our proposed metric can be used in all seven tasks with the same standard. We adopt two major evaluation metrics, namely semantic consistency $SC$ and perceptive quality $PQ$. These two metrics measure the quality of the generated images from two aspects. The semantic consistency measures how well the generated image is aligned with the condition $X = [c_1, c_2, \cdots]$. Specifically, we define

| Condition 1 | Condition 2 | Condition 3 | SC rating |
|---|---|---|---|
| Inconsistent | Any | Any | 0 |
| Any | Inconsistent | Any | 0 |
| Any | Any | Inconsistent | 0 |
| Partially Consistent | Any | Any | 0.5 |
| Any | Partially Consistent | Any | 0.5 |
| Any | Any | Partially Consistent | 0.5 |
| Mostly Consistent | Mostly Consistent | Mostly Consistent | 1.0 |

| Subjects in image | Artifacts | Unusual sense | PQ rating |
|---|---|---|---|
| Unrecognizable | Any | Any | 0 |
| Any | Serious | Any | 0 |
| Recognizable | Moderate | Any | 0.5 |
| Recognizable | Any | Moderate | 0.5 |
| Recognizable | Little/None | Little/None | 1.0 |

Table 1: Rating guideline for computing the SC and PQ score. Detail in subsection A.2.

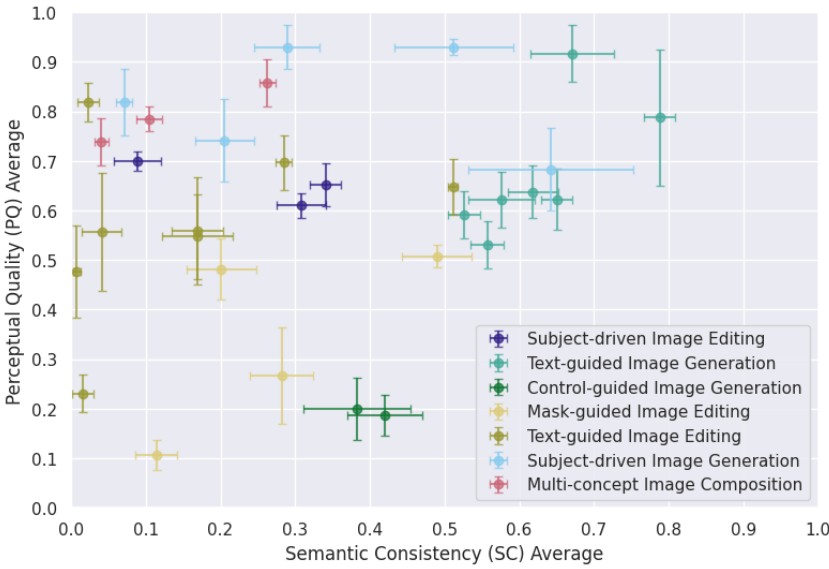

Figure 4: Model performance and standard deviation in each task.

the semantic consistency score as:

$$SC(y, X) = min\{g(y, c_1), g(y, c_2), \cdots, g(y, c_k)\} \qquad (1)$$

where $g$ is a modularized function to compute the consistency between $y$ and a single condition $c$. We set $g(y, c_1) \in [0, 0.5, 1]$, where 0 means inconsistent, 0.5 means partially consistent and 1 means fully consistent. With this formulation, as long as the output is inconsistent with any of the conditions, the evaluation score should become zero. Otherwise, the aggregation function will pick the lowest consistency score from all the conditions. On the other hand, perceptive quality measures the image quality, i.e. whether the image contains artifacts, is blurry, or has an unnatural sense. we set perceptive quality $PQ \in [0, 0.5, 1]$, where 0 means extremely poor quality, 0.5 means the image has an acceptable quality and 1 means high quality. In these experiments, each model is rated individually. We train human raters to estimate the $g$ function and $PQ$ function with comprehensive guidelines in Table 1. We derive $O = \sqrt{SC \times PQ}$ as the overall rating of a model. One benefit of using geometric mean as the design choice is that the rating is penalized when one of the aspect scores is too low. We further studied the configuration in section 5.

| Task | Data Source | Inference Dataset size |
|------|-------------|------------------------|
| Text-guided Image Generation | DrawBench (Saharia et al., 2022) 
 DiffusionDB (Wang et al., 2022) 
 ABC-6K (Feng et al., 2023) 
 Ours | 197 |
| Mask-guided Image Editing | MagicBrush (Zhang et al., 2023) 
 Ours | 179 |
| Text-guided Image Editing | MagicBrush (Zhang et al., 2023) 
 Ours | 179 |
| Subject-driven Image Generation | SuTI (Chen et al., 2023) 
 Ours | 150 |
| Multi-concept Image Composition | CustomDiffusion (Kumari et al., 2023) 
 Ours | 102 |
| Subject-driven Image Editing | DreamEditBench (Li et al., 2023b) | 154 |
| Control-guided Image Generation | HuggingFace community | 150 |

Table 2: All the human evaluation datasets from seven core tasks.

## 4.2 DATASET AND AVAILABLE MODELS

We present a standardized dataset for each type of task. The information of the datasets is shown in Table 2. Some models have different standards of inputs in one task. For example, in the text-guided image editing task, DiffEdit is global description-guided while InstructPix2Pix is instruction-guided. We manually created the equivalent meaning prompts for both methods so they can be aligned. All datasets contain a huge variety of test cases to mimic the diversity in real-life situations. We hosted all of our datasets on the HuggingFace dataset for easy access and maintenance. Here we demonstrate all the evaluated models in Table 3.

## 5 EXPERIMENTAL RESULTS

**Experiment Setup.** All the models either used the default setting from the official implementation or the setting suggested in HuggingFace documentation (von Platen et al., 2022). We disabled negative prompts and any prompt engineering tricks to ensure a fair comparison. We conducted human evaluation by recruiting participants from Prolific to rate the images, and our own researchers also took part in the image rating process. We assigned 3 raters for each model and computed the SC score, PQ score, and Overall human score. Then we computed the Fleiss kappa (Fleiss & Cohen, 1973) for each mentioned score. We also computed Krippendorff's Alpha (Krippendorff, 2011), which is expected to yield a higher value than Fleiss kappa. This distinction arises from the nature of the rating categories, with Fleiss' Kappa assuming nominal categories and Krippendorff's Alpha accommodating ordinal inputs. Both Fleiss' Kappa and Krippendorff's Alpha are bounded within the range of [-1, 1], where the value $> 0$ indicates an agreement and closer proximity to 1 indicates a higher degree of agreement. The design choices are explained in subsection 5.2 and Appendix A.6.

**Results.** In Figure 4, we present an overview of the model performance across various tasks. Our findings indicate that the performance of the current models is generally underwhelming, with the exception being Text-guided Image Generation and Subject-driven Image Generation, which have models reaching higher than 0.6 on both SC and PQ averages. The detailed report on each model's performance is shown on Table 4. We noticed that the overall automated metrics' correlation with the SC score and PQ score in each task is below 0.2 except subject-driven image editing task. Metric values are at Table 5 and Table 6.

### 5.1 DISCOVERY AND INSIGHTS

**Text-Guided Image Generation.** We observe that all models are able to generate high-quality images. Regarding semantic consistency, all models have a good understanding of the general prompts, while Stable Diffusion XL is better at understanding complex prompts. For example, it exhibits a high degree of accuracy and detail on the prompt "A panda making latte art." while other models

| Model | #Params | Runtime | Keywords for technical detail |
|---|---|---|---|
| *Text-to-Image Generation* | | | |
| Dalle-3 (openai, 2023) | - | - | Recaptioned training data |
| MidJourney (midjourney, 2023) | - | - | - |
| Dalle-2 (Ramesh et al., 2022) | 3.5B | 10s | unCLIP, two-stage |
| Stable Diffusion (Rombach et al., 2022) | 0.8B | 3s | Latent Diffusion |
| DeepFloydIF (deep floyd.ai, 2023) | 4.3B | 37s | Cascaded Pixel Diffusion |
| OpenJourney (openjourney.ai, 2023) | 0.8B | 3s | SD, Midjourney data |
| Stable Diffusion XL (stability.ai, 2023) | 2.3B | 11s | Stable Diffusion, X-Large |
| *Mask-guided Image Editing* | | | |
| BlendedDiffusion (Avrahami et al., 2022) | 0.5B | 57s | Noise Blending, DDPM+CLIP |
| GLIDE (Nichol et al., 2022) | 3.5B | 19s | CLIP, Diffusion |
| SD-Inpaint (runwayml, 2023) | 1.1B | 11s | SD, Inpainting training |
| SDXL-Inpaint (stability.ai, 2023) | 2.7B | 36s | SDXL, Inpainting training |
| *Text-guided Image Editing* | | | |
| SDEdit (Meng et al., 2021) | 1.3B | 13s | SDE Prior |
| Text2Live (Bar-Tal et al., 2022) | 3.1M | 36s | Zero-shot, Edit layer |
| DiffEdit (Couairon et al., 2022) | 1.3B | 29s | Mask estimation |
| Cycle Diffusion (Wu & la Torre, 2023) | 1.1B | 9s | DPM-Encoder, Zero-shot |
| Prompt-to-Prompt (Mokady et al., 2022) | 1.1B | 2m | Cross-Attention |
| Pix2PixZero (Parmar et al., 2023) | 1.1B | 21s | Cross-Attention, Zero Prompt |
| InstructPix2Pix (Brooks et al., 2023) | 1.1B | 11s | SD, synthetic P2P data |
| MagicBrush (Zhang et al., 2023) | 1.1B | 7s | SD, MagicBrush data |
| *Subject-driven Image Generation* | | | |
| Textual Inversion (Gal et al., 2022) | 1.1B | 15m | Word embedding tuning |
| DreamBooth (Ruiz et al., 2023) | 1.1B | 10m | Finetuning with preservation loss |
| DreamBooth-Lora (Hu et al., 2021) | 1.1B | 8m | DreamBooth + Low-Rank Adaptation |
| SuTI (Chen et al., 2023) | 2.5B | 30s | In-context + Apprenticeship learning |
| BLIP-Diffusion (Li et al., 2023a) | 1.1B | 8s | Pretrained encoder, Zero-shot |
| *Subject-driven Image Editing* | | | |
| DreamEdit (Li et al., 2023b) | 1.1B | 8m | Dreambooth + Region proposal |
| PhotoSwap (Gu et al., 2023) | 1.1B | 7m | Dreambooth + Cross-Attention |
| BLIP-Diffusion (Li et al., 2023a) | 1.1B | 18s | Pretrained encoder, Zero-shot |
| *Multi-concept Image Composition* | | | |
| CustomDiffusion (Kumari et al., 2023) | 1.1B | 19m | Cross-attention updating |
| DreamBooth (Ruiz et al., 2023) | 1.1B | 11m | Finetuning with preservation loss |
| TextualInversion (Gal et al., 2022) | 1.1B | 32m | Word embedding tuning |
| *Control-guided Image Generation* | | | |
| ControlNet (Zhang & Agrawala, 2023) | 1.4B | 8s | Zero convolution + Frozen model |
| UniControl (Qin et al., 2023) | 1.4B | 23s | Multi-task pretraining, Zero-shot |

Table 3: Overview of all the evaluated models and their parameter size and runtime. The models are listed in chronological order. The number of parameters and runtime for Dalle-3 and MidJourney are unknown. MidJourney is not opensource and does not have a whitepaper in technical detail.

often misunderstood the prompt as "panda latte art". DALLE-3 and Midjourney understand most of the complex prompts than other models, and DALLE-3 is slightly better on conflicting prompts.

**Mask-guided Image Editing.** We observed the outputs commonly contain obvious artifacts in the masked region boundaries for Stable Diffusion and GLIDE. While Blended Diffusion and Stable Diffusion XL do not suffer from the same issue, they often produce unrecognizable outputs. Stable Diffusion XL obtains the best results but the overall model performance is still far from satisfactory. Another common issue is that the filled regions can hardly harmonize with the background.

**Text-guided Image Editing.** One key requirement is to edit the image precisely and keep the background untouched. This requirement is indeed challenging because the network has to understand of editing region from the semantic inputs. We discovered that Prompt-to-Prompt, Pix2PixZero, and SDEdit, despite generating high-quality images, often result in completely different backgrounds. We also spotted that in many cases Text2Live will simply return the input as output, this phenomenon

| Model | LPIPS ↓ | CLIP ↑ | $SC_{Avg}$ | $PQ_{Avg}$ | Overall | $Fleiss_{\bar{O}}$ | $Kd_{\bar{O}}$ |
|---|---|---|---|---|---|---|---|
| Text-guided Image Generation | | | | | | | |
| Dalle-3 | N/A | 0.2697 | 0.79±0.02 | 0.79±0.14 | **0.76±0.08** | 0.19 | 0.34 |
| Midjourney | N/A | 0.2839 | 0.67±0.06 | 0.92±0.06 | 0.73±0.07 | 0.34 | 0.51 |
| DeepFloydIF | N/A | 0.2814 | 0.65±0.02 | 0.62±0.06 | 0.59±0.02 | 0.32 | 0.51 |
| Stable Diffusion XL | N/A | 0.2886 | 0.62±0.03 | 0.64±0.05 | 0.59±0.03 | 0.37 | 0.61 |
| Dalle-2 | N/A | 0.2712 | 0.58±0.04 | 0.62±0.06 | 0.54±0.04 | 0.27 | 0.40 |
| OpenJourney | N/A | 0.2814 | 0.53±0.02 | 0.59±0.05 | 0.50±0.02 | 0.30 | 0.47 |
| Stable Diffusion 2.1 | N/A | 0.2899 | 0.56±0.02 | 0.53±0.05 | 0.50±0.03 | 0.38 | 0.50 |
| Mask-guided Image Editing | | | | | | | |
| SDXL-Inpainting | 0.15 | 0.2729 | 0.49±0.05 | 0.51±0.02 | **0.37±0.05** | 0.50 | 0.72 |
| SD-Inpainting | 0.21 | 0.2676 | 0.28±0.04 | 0.27±0.10 | 0.17±0.07 | 0.31 | 0.49 |
| GLIDE | 0.18 | 0.2578 | 0.20±0.05 | 0.48±0.06 | 0.16±0.05 | 0.33 | 0.56 |
| BlendedDiffusion | 0.33 | 0.2594 | 0.12±0.03 | 0.11±0.03 | 0.05±0.02 | 0.36 | 0.44 |
| Text-guided Image Editing | | | | | | | |
| MagicBrush | 0.22 | 0.2675 | 0.51±0.01 | 0.65±0.06 | **0.47±0.02** | 0.44 | 0.67 |
| InstructPix2Pix | 0.32 | 0.2616 | 0.29±0.01 | 0.70±0.06 | 0.27±0.02 | 0.55 | 0.74 |
| Prompt-to-prompt | 0.40 | 0.2674 | 0.17±0.05 | 0.55±0.09 | 0.15±0.06 | 0.36 | 0.53 |
| CycleDiffusion | 0.28 | 0.2692 | 0.17±0.03 | 0.56±0.11 | 0.14±0.04 | 0.41 | 0.63 |
| SDEdit | 0.61 | 0.2872 | 0.04±0.03 | 0.56±0.12 | 0.04±0.03 | 0.13 | 0.13 |
| Text2Live | 0.17 | 0.2628 | 0.02±0.01 | 0.82±0.04 | 0.02±0.02 | 0.10 | 0.17 |
| DiffEdit | 0.22 | 0.2425 | 0.02±0.01 | 0.23±0.04 | 0.01±0.01 | 0.24 | 0.24 |
| Pix2PixZero | 0.60 | 0.2510 | 0.01±0.00 | 0.48±0.09 | 0.01±0.01 | 0.37 | 0.37 |
| Subject-driven Image Generation | | | | | | | |
| SuTI | 0.77 | 0.2895 | 0.64±0.11 | 0.68±0.08 | **0.58±0.12** | 0.20 | 0.39 |
| DreamBooth | 0.77 | 0.2847 | 0.51±0.08 | 0.93±0.02 | 0.55±0.11 | 0.37 | 0.60 |
| BLIP-Diffusion | 0.77 | 0.2729 | 0.29±0.04 | 0.93±0.04 | 0.35±0.06 | 0.22 | 0.39 |
| TextualInversion | 0.81 | 0.2680 | 0.21±0.04 | 0.74±0.08 | 0.21±0.05 | 0.35 | 0.52 |
| DreamBooth-Lora | 0.82 | 0.2988 | 0.07±0.01 | 0.82±0.07 | 0.09±0.01 | 0.29 | 0.37 |
| Subject-driven Image Editing | | | | | | | |
| PhotoSwap | 0.34 | 0.2846 | 0.34±0.02 | 0.65±0.04 | **0.36±0.02** | 0.35 | 0.46 |
| DreamEdit | 0.22 | 0.2855 | 0.31±0.03 | 0.61±0.03 | 0.32±0.03 | 0.33 | 0.52 |
| BLIP-Diffusion | 0.25 | 0.2901 | 0.09±0.03 | 0.70±0.02 | 0.09±0.03 | 0.41 | 0.47 |
| Multi-concept Image Composition | | | | | | | |
| CustomDiffusion | 0.79 | 0.2929 | 0.26±0.01 | 0.86±0.05 | **0.29±0.01** | 0.73 | 0.88 |
| DreamBooth | 0.78 | 0.2993 | 0.11±0.02 | 0.78±0.02 | 0.13±0.02 | 0.61 | 0.71 |
| TextualInversion | 0.80 | 0.2548 | 0.04±0.01 | 0.74±0.05 | 0.05±0.01 | 0.62 | 0.77 |
| Control-guided Image Generation | | | | | | | |
| ControlNet | 0.80 | 0.2555 | 0.42±0.05 | 0.19±0.04 | **0.23±0.04** | 0.37 | 0.57 |
| UniControl | 0.82 | 0.2604 | 0.38±0.07 | 0.20±0.06 | 0.23±0.07 | 0.36 | 0.58 |

Table 4: All the evaluated models from seven core tasks. Overall is the average of all $\sqrt{SC \times PQ}$. $Fleiss_{\bar{O}}$ and $Kd_{\bar{O}}$, denoting Fleiss' Kappa and Krippendorff's alpha for the overall average, respectively. We have more automated metric results in Appendix Table 5 and correlations in Table 6.

also occasionally happened in other models. For paper claims, our evaluation ranking aligns with the findings from CycleDiffsuion, InstructPix2Pix, MagicBrush, Prompt-to-Prompt, and DiffEdit. We found our evaluation ranking does not align with Pix2PixZero, since their paper only tested on word-swapping examples, which is not able to generalize to more complex edits.

**Subject-driven Image Generation.** Our evaluation results largely align with DreamBooth, BLIP-Diffusion, and SuTI findings. Specifically, Textual inversion struggles to maintain target subject features. DreamBooth can imitate subjects based on images but occasionally resorts to copying learned images. DreamBooth-Lora struggles to generate desired subjects but can follow context prompts. BLIP-Diffusion can mimic target subject features but struggles with details. SuTI maintains high consistency with desired subjects and context, with tolerable artifacts in some cases.

**Multi-concept Image Composition.** Our evaluations validate that CustomDiffusion is consistently better than the other two models. However, while it learns the given multiple subjects' features

better, it could fail to follow the prompts in many cases, especially on actions and positional words. In contrast, DreamBooth learns the correct subjects in some cases, and TextualInversion rarely learns the correct subjects. Interestingly, in some cases where DreamBooth does not learn the correct subjects, it could still follow the prompts correctly.

**Subject-driven Image Editing.** It is essential to modify the subject from the source to the target without causing excess changes to the background. Human evaluation is also conducted in DreamEdit for the comparison between Photoswap and DreamEdit, but our rankings differ due to varying evaluation criteria. PhotoSwap can adapt to the target subject from the source naturally in most cases but rarely preserves the background well. DreamEdit maintains the context in most cases but sometimes leaves observable distortions at the edge of contextualization. BLIP-Diffusion fails the adaptation most of the time, compromising for a more realistic generation.

**Control-guided Image Generation.** Our evaluation shows there is no significant difference between the two models in both automatic metrics and human evaluation metrics. While UniControl also reported that there is no significant difference in automatic metrics, our human evaluation results do not align. This can be due to the different evaluation standards and aspects. Nevertheless, it has come to our attention that neither of these models demonstrates a high level of robustness. Scratches often appeared on the generated image.

## 5.2 ABLATION STUDY

**Method of overall human score computation.** We set the overall score of the model as the geometric mean of SC and PR score (i.e. $O = \sqrt{SC \times PQ}$). But we also explored the weighted sum setting $O = \alpha \times SC + \beta \times PQ$, where both $\alpha$ and $\beta$ are in [0, 1]. We experimented and found that the weighted sum setting yields a different ranking in the models. Take the text-guided image editing task as an example, as in Appendix Table 7, Text2Live outperforms CycleDiffusion in the weighted sum setting even though we found that CycleDiffusion performs better in the human examination. We investigated and found that a majority of results in Text2Live simply return the input as output (in that case SC=0 and PQ=1). We tried adjusting the weightings but it still failed to reflect the actual performance of the model. Thus we decided to use the geometric mean setting to penalize the models.

**Design choice of human evaluation metric range.** When it comes to human evaluation on a massive scale, it's essential to create a system that's easy to understand and quick to use. In this investigation, we undertake an exploration into how different settings in this evaluation method can affect the results, showing in Appendix Table 8. Initially, our approach entailed the utilization of a range encompassing [0, 0.5, 1, 2] for both the Semantic Consistency (SC) score and the Perception Quality (PQ) score, where 2 means the image is perfect. However, this configuration yielded suboptimal results in the Fleiss Kappa. Subsequently, an alternative configuration was employed, narrowing the range to [0, 1] for both the SC and PQ scores. This adjustment, while accommodating a binary classification, was observed to yield values that were overly polarized and extreme. To find the right balance between keeping values in a reasonable range and making sure the evaluation method is reliable, we resolved to define a range of [0, 0.5, 1] while providing explicit and unambiguous guidelines.

## CONCLUSION

In this paper, we propose `ImagenHub` as a continuous effort to unify all efforts in conditional image generation into a library, easing access to these models. We standardize the dataset and evaluation of these models to build our `ImagenHub` Leaderboard. We hope this leaderboard can provide a more reproducible and fair environment for researchers to visualize progress in this field. A limitation of this work is the reliance on human raters, which is expensive and time-consuming. In the future, we plan to develop more generic automatic evaluation methods that approximate human ratings, helping people develop better models.

ETHICS STATEMENT

Our work aims to benefit the broader research community by providing a standardized framework for evaluating conditional image generation models. For all the benchmarks, we are committed to the ethical use of them. The datasets used in our work are either publicly available or have been collected and curated exercising the utmost respect for privacy and consent. We will maintain a leaderboard to track latest models and encourage open collaboration and discussion in the field. In human evaluation, we followed the minimum hourly wage of $11. We also ensure that no personal information is collected and no offensive content is presented during human evaluations.

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

# A APPENDIX

## A.1 MORE METRICS RESULTS

| Model | LPIPS ↓ | DINO ↑ | CLIP-I ↑ | DreamSim ↓ | FID ↓ | KID ↓ | CLIP ↑ |
|---|---|---|---|---|---|---|---|
| Text-guided Image Generation | | | | | | | |
| Dalle-3 | N/A | N/A | N/A | N/A | N/A | N/A | 0.2697 |
| MidJourney | N/A | N/A | N/A | N/A | N/A | N/A | 0.2839 |
| DeepFloydIF | N/A | N/A | N/A | N/A | N/A | N/A | 0.2814 |
| Stable Diffusion XL | N/A | N/A | N/A | N/A | N/A | N/A | 0.2886 |
| Dalle-2 | N/A | N/A | N/A | N/A | N/A | N/A | 0.2712 |
| OpenJourney | N/A | N/A | N/A | N/A | N/A | N/A | 0.2814 |
| Stable Diffusion | N/A | N/A | N/A | N/A | N/A | N/A | 0.2899 |
| Mask-guided Image Editing | | | | | | | |
| SDXL-Inpainting | 0.15 | 0.90 | 0.94 | 0.15 | 59.04 | -0.0033 | 0.2729 |
| SD-Inpainting | 0.21 | 0.85 | 0.91 | 0.20 | 86.23 | -0.0008 | 0.2676 |
| GLIDE | 0.18 | 0.91 | 0.94 | 0.14 | 61.71 | -0.0028 | 0.2578 |
| BlendedDiffusion | 0.33 | 0.80 | 0.87 | 0.26 | 102.73 | -0.0012 | 0.2594 |
| Text-guided Image Editing | | | | | | | |
| MagicBrush | 0.22 | 0.88 | 0.92 | 0.19 | 77.81 | -0.0027 | 0.2675 |
| InstructPix2Pix | 0.32 | 0.77 | 0.86 | 0.30 | 100.30 | -0.0014 | 0.2616 |
| Prompt-to-prompt | 0.40 | 0.68 | 0.81 | 0.36 | 137.68 | 0.0003 | 0.2674 |
| CycleDiffusion | 0.28 | 0.79 | 0.88 | 0.27 | 108.29 | -0.0008 | 0.2692 |
| SDEdit | 0.61 | 0.57 | 0.76 | 0.50 | 156.12 | 0.0025 | 0.2872 |
| Text2Live | 0.17 | 0.87 | 0.92 | 0.18 | 68.33 | -0.0028 | 0.2628 |
| DiffEdit | 0.22 | 0.78 | 0.85 | 0.31 | 121.79 | 0.0069 | 0.2425 |
| Pix2PixZero | 0.60 | 0.53 | 0.76 | 0.52 | 161.64 | 0.0072 | 0.2510 |
| Subject-driven Image Generation | | | | | | | |
| SuTI | 0.77 | 0.69 | 0.81 | 0.32 | 164.97 | 0.0111 | 0.2895 |
| DreamBooth | 0.77 | 0.61 | 0.80 | 0.40 | 173.71 | 0.0057 | 0.2847 |
| BLIP-Diffusion | 0.77 | 0.63 | 0.82 | 0.36 | 166.53 | 0.0102 | 0.2729 |
| TextualInversion | 0.81 | 0.37 | 0.65 | 0.61 | 229.18 | 0.0122 | 0.2680 |
| DreamBooth-Lora | 0.82 | 0.36 | 0.70 | 0.61 | 234.95 | 0.0125 | 0.2988 |
| Subject-driven Image Editing | | | | | | | |
| PhotoSwap | 0.34 | 0.65 | 0.83 | 0.37 | 130.81 | 0.0009 | 0.2846 |
| DreamEdit | 0.22 | 0.74 | 0.87 | 0.28 | 118.95 | 0.0011 | 0.2855 |
| BLIP-Diffusion | 0.25 | 0.80 | 0.89 | 0.23 | 102.17 | 0.0007 | 0.2901 |
| Multi-concept Image Composition | | | | | | | |
| CustomDiffusion | 0.79 | 0.50 | 0.70 | 0.55 | 207.43 | 0.0341 | 0.2929 |
| DreamBooth | 0.78 | 0.39 | 0.64 | 0.66 | 257.60 | 0.0533 | 0.2993 |
| TextualInversion | 0.80 | 0.44 | 0.66 | 0.65 | 226.25 | 0.0491 | 0.2548 |
| Control-guided Image Generation | | | | | | | |
| ControlNet | 0.80 | 0.43 | 0.70 | 0.56 | 219.29 | 0.0330 | 0.2555 |
| UniControl | 0.82 | 0.42 | 0.68 | 0.58 | 222.57 | 0.0192 | 0.2604 |

Table 5: All automated metrics for the evaluated models based on our benchmark dataset.

| Task | PQ ↑ | | | | SC ↑ |
|---|---|---|---|---|---|
| | corr(LPIPS) | corr(DINO) | corr(CLIP-I) | corr(DreamSim) | corr(CLIP) |
| Text-guided Image Generation | N/A | N/A | N/A | N/A | -0.0819 |
| Mask-guided Image Editing | -0.0502 | -0.0421 | -0.0308 | -0.0294 | -0.0257 |
| Text-guided Image Editing | 0.0439 | 0.0449 | 0.0765 | 0.0432 | 0.0437 |
| Subject-driven Image Generation | 0.0905 | 0.1737 | 0.1204 | 0.0943 | -0.0117 |
| Subject-driven Image Editing | 0.2417 | 0.1690 | 0.3770 | 0.3846 | -0.0443 |
| Multi-concept Image Composition | -0.1018 | -0.1657 | -0.0965 | 0.0140 | -0.0107 |
| Control-guided Image Generation | 0.1864 | 0.1971 | -0.0985 | -0.0045 | -0.2056 |

Table 6: Metrics correlation. we inverted the signs for metric that is the lower the better. The computable metrics all hold different assumptions which only contribute part of the aspects in human evaluation. This makes a low correlation with human evaluations.

| Setting | $O = \sqrt{SC \times PQ}$ | | $O = 0.5SC + 0.5PQ$ | | $O = 0.7SC + 0.3PQ$ | |
|---|---|---|---|---|---|---|
| | $O_{Sum}$ | $O_{Avg}$ | $O_{Sum}$ | $O_{Avg}$ | $O_{Sum}$ | $O_{Avg}$ |
| MagicBrush | 83.51 | 0.47 | 103.75 | 0.58 | 98.85 | 0.55 |
| CycleDiffusion | 24.89 | 0.14 | 65.25 | 0.36 | 51.28 | 0.29 |
| DiffEdit | 1.71 | 0.01 | 22.08 | 0.12 | 14.38 | 0.08 |
| Text2Live | 4.08 | 0.02 | 75.25 | 0.42 | 46.75 | 0.26 |

Table 7: Comparison on overall human score computation setting.

| Setting | categories = [0, 0.5, 1] | | | categories = [0, 0.5, 1, 2] | | | categories = [0, 1] | | |
|---|---|---|---|---|---|---|---|---|---|
| | $O_{Avg}$ | $Fleiss_{\bar{O}}$ | $Kd_{\bar{O}}$ | $O_{Avg}$ | $Fleiss_{\bar{O}}$ | $Kd_{\bar{O}}$ | $O_{Avg}$ | $Fleiss_{\bar{O}}$ | $Kd_{\bar{O}}$ |
| BLIP-Diffusion | 0.13 | 0.41 | 0.47 | 0.09 | 0.29 | 0.50 | 0.05 | 0.40 | 0.40 |
| DreamEdit | 0.50 | 0.33 | 0.52 | 0.32 | 0.26 | 0.55 | 0.21 | 0.38 | 0.38 |
| PhotoSwap | 0.62 | 0.35 | 0.46 | 0.36 | 0.25 | 0.50 | 0.29 | 0.37 | 0.37 |

Table 8: Ablation study to understand the impact of the granularity of SC and PQ.

## A.2 HUMAN EVALUATION GUIDELINE

To standardize the conduction of a rigorous human evaluation, we stipulate the criteria for each measurement as follows:

- Semantic Consistency (SC), score in range [0, 0.5, 1]. It measures the level that the generated image is coherent in terms of the condition provided (i.e. Prompts, Subject Token, etc.).
- Perceptual Quality (PQ), score in range [0, 0.5, 1]. It measures the level at which the generated image is visually convincing and gives off a natural sense.

Meaning of Semantic Consistency (SC) score:

- SC=0 : Image not following one or more of the conditions at all (e.g. not following the prompt at all, different background in editing task, wrong subject in subject-driven task, etc.)
- SC=0.5 : all the conditions are partly following the requirements.
- SC=1 : The rater agrees that the overall idea is correct.

Meaning of Perceptual Quality (PQ) score:

- PQ=0 : The rater spotted obvious distortion or artifacts at first glance and those distorts make the objects unrecognizable.
- PQ=0.5 : The rater found out the image gives off an unnatural sense. Or the rater spotted spotted some minor artifacts and the objects are still recognizable.
- PQ=1 : The rater agrees that the resulting image looks genuine.

Raters have to strictly adhere to Table 9 when rating.

| Condition 1 | Condition 2 (if applicable) | Condition 3 (if applicable) | SC rating |
|---|---|---|---|
| no following at all | Any | Any | 0 |
| Any | no following at all | Any | 0 |
| Any | Any | no following at all | 0 |
| following some part | following some or most part | following some or most part | 0.5 |
| following some or most part | following some part | following some or most part | 0.5 |
| following some part or more | following some or most part | following some part | 0.5 |
| following most part | following most part | following most part | 1 |

| Objects in image | Artifacts | Unusual sense | PQ rating |
|---|---|---|---|
| Unrecognizable | serious | Any | 0 |
| Recognizable | some | Any | 0.5 |
| Recognizable | Any | some | 0.5 |
| Recognizable | none | little or None | 1 |

Table 9: Rating guide checklist table.

Artifacts and Unusual sense, respectively, are:

- Distortion, watermark, scratches, blurred faces, unusual body parts, subjects not harmonized
- wrong sense of distance (subject too big or too small compared to others), wrong shadow, wrong lighting, etc.

## A.3 RATING EXAMPLES

There are some examples when evaluating:

**Text-to-Image Generation.**

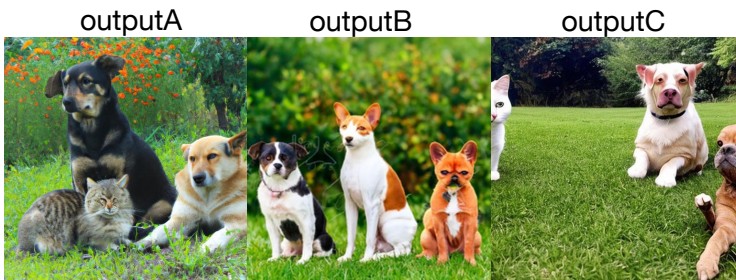

"prompt": "One cat and two dogs sitting on the grass.",
"category": "Counting".

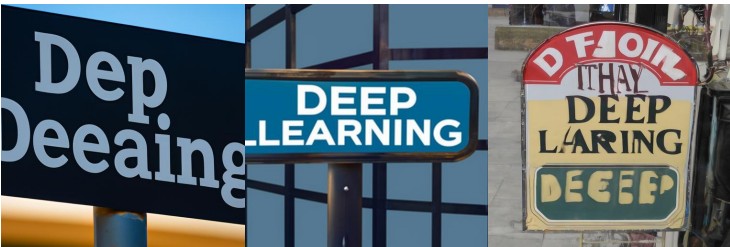

"prompt": "A sign that says 'Deep Learning'.",
"category": "Text"

Figure 5: Rating examples on Text-to-Image Generation task.

- OutputA$_1$: `[1, 0.5]`. SC=1: Prompt perfectly align. PR=0.5: Minor distortion on the cat's facial features.
- OutputB$_1$: `[0.5, 0.5]`. SC=0.5: 3 dogs appeared instead of 1 cat and 2 dogs. PR=0.5: Minor distortion was found on the animal's face and the watermark.
- OutputC$_1$: `[1, 0.5]`. SC=1: The prompt match perfectly with the image. PR=0.5: Minor distortion on the dog's facial features.
- OutputA$_2$: `[0.5, 1]`. SC=0.5: A sign appeared, but failed to spell the word. PR=1: The image look generally real but with some lighting issues.
- OutputB$_2$: `[0.5, 0.5]`. SC=0.5: A sign appeared, but failed to spell the word. PR=0.5: The background looks so unnatural.
- OutputC$_2$: `[0.5, 0]`. SC=0.5: A sign appeared, but failed to spell the word. PR=0: Heavy distortion on both text and strong artifacts in the background.

**Mask-guided Image Editing.**

input          mask          outputA          outputB

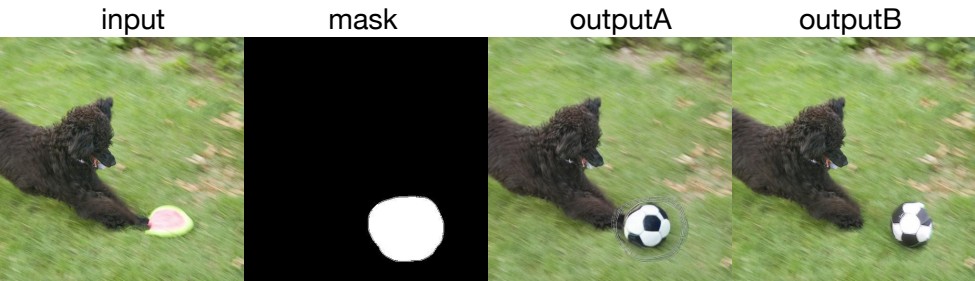

"source_global_caption": "A small black dog playing with a frisbee.",
"instruction": "turn the frisbee into a soccer ball",
"target_global_caption": "A small black dog playing with a soccer ball."

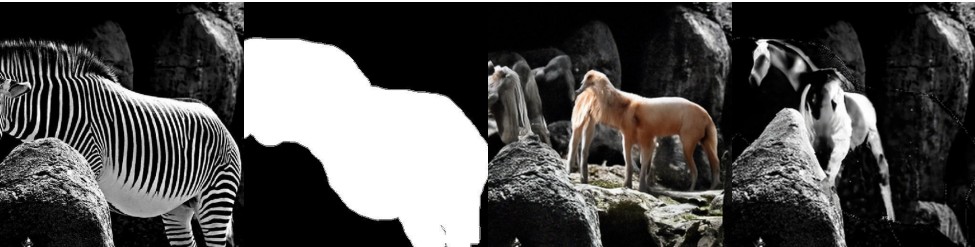

"source_global_caption": "A Zebra standing in between a group of large rocks",
"instruction": "Make the zebra a regular horse.",
"target_global_caption": "A horse standing in between a group of large rocks"

Figure 6: Rating examples on Mask-guided Image Editing task.

- OutputA$_1$: `[1, 0.5]`. SC=1: Clearly generate the soccer. PR=0.5: The edit region does not blend well with the context.
- OutputB$_1$: `[1, 1]`. SC=1: Successfully add a soccer. PR=1: The soccer is naturally blended with the context.
- OutputA$_2$: `[0, 0]`. SC=0: Generated content can not be regarded as horse. PR=0: The middle left part is not natural.
- OutputB$_2$: `[0.5, 0]`. SC=0.5: The object is horse-like but not good. PR=0: The whole image is not natural.

**Text-guided Image Editing.**

|  input  |  outputA  |  outputB  |

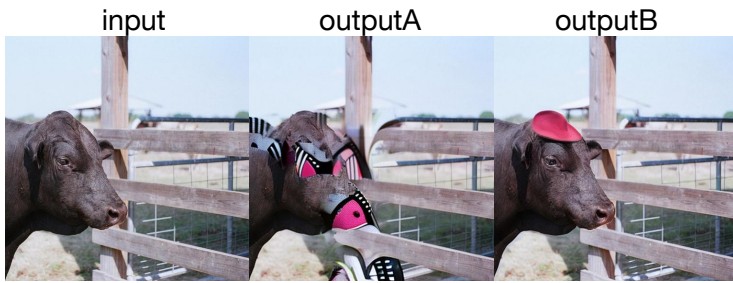

"source_global_caption": "A bull is on a farm walking around a pen.",
"instruction": "Have the cow wear a hat.",
"target_global_caption": "A stylish cow wearing a hat walks around a pen on a farm."

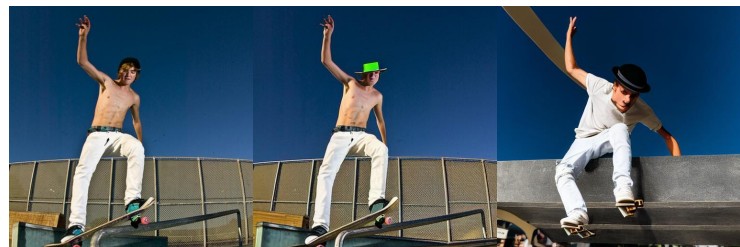

"source_global_caption": "A skateboarder is doing a trick on a hand rail.",
"instruction": "What if the man had a hat?",
"target_global_caption": "A skateboarder with a hat is doing a trick on a hand rail."

Figure 7: Rating examples on Text-guided Image Editing task.

- $OutputA_1$: `[0, 0]`. SC=0: The image does not follow the instruction at all. PR=0: Heavy distortion on the cow's facial features.
- $OutputB_1$: `[1, 1]`. SC=1: The hat gives a nice specular reflection. PR=1: It looks real.
- $OutputA_2$: `[0.5, 0.5]`. SC=0.5: The hat exists but does not suit well. PR=0.5: The important object look distorted.
- $OutputB_2$: `[0, 0]`. SC=0: The background completely changed. PR=0: The whole image look distorted.

**Subject-driven Image Generation.**

input          outputA          outputB

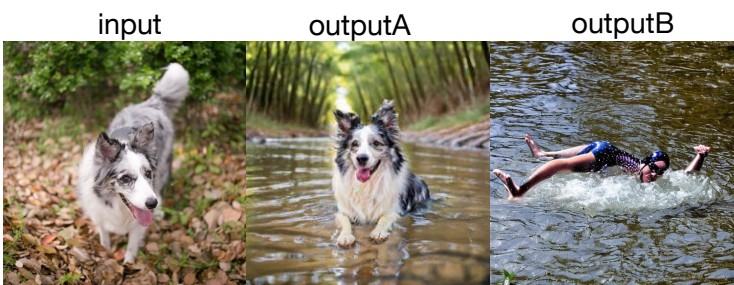

"subject_id": 16,
"subject": "dog8",
"prompt": "A <token> dog swimming in a river."

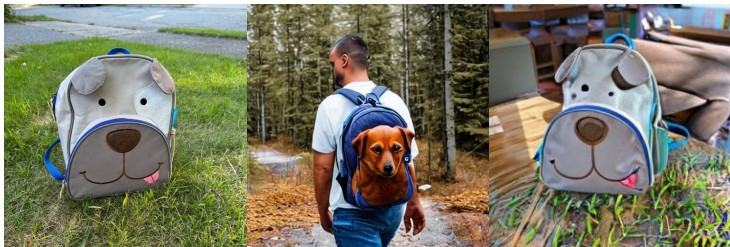

"subject_id": 1,
"subject": "backpack_dog",
"prompt": "a man carrying <token> backpack dog."

Figure 8: Rating examples on Subject-driven Image Generation task.

- Output$A_1$: [1, 1]. SC=1: The output does match the subject and prompt. PR=1: It looks real.
- Output$B_1$: [0, 0]. SC=0: The subject dog missing. PR=0: Serious unusual body.
- Output$A_2$: [0, 0.5]. SC=0: The output does not match the subject. PR=0.5: Some unusual sense on the backpack.
- Output$B_2$: [0, 0.5]. SC=0: The output does not match the prompt. PR=0.5: Some unusual sense on the desk with the grass.

**Multi-concept Image Composition.**

input                    outputA       outputB

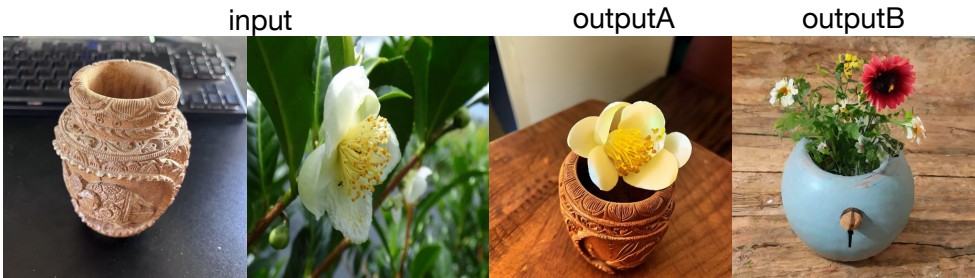

"target_caption": "flower in the wooden pot on a table."

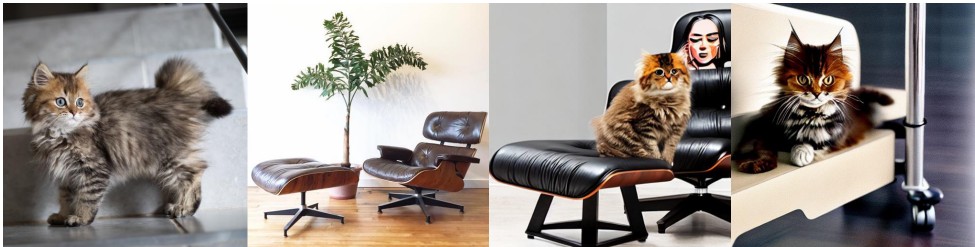

"target_caption": "chair with a screaming cat sitting on it."

Figure 9: Rating examples on Multi-concept Image Composition task.

- OutputA$_1$: [1, 1]. SC=1: Both subjects accurately represent the intended subjects and the prompt actions match. PR=1: In general the image is natural and as real.

- OutputB$_1$: [0, 1]. SC=0: Subjects are not correct even though prompt action matches. PR=1: The image looks real.

- OutputA$_2$: [0.5, 0.5]. SC=0.5: Both subjects accurately represent the intended subjects and but the prompt action doesn't match. PR=0.5: There is a human face on the chair so the important subject chair looks unrealistic, but do not strongly detract from the image's overall appearance.

- OutputB$_2$: [0, 0.5]. SC=0: Subject chair missing. PR=0.5: Minor distortion on the cat body and background.

**Subject-driven Image Editing.**

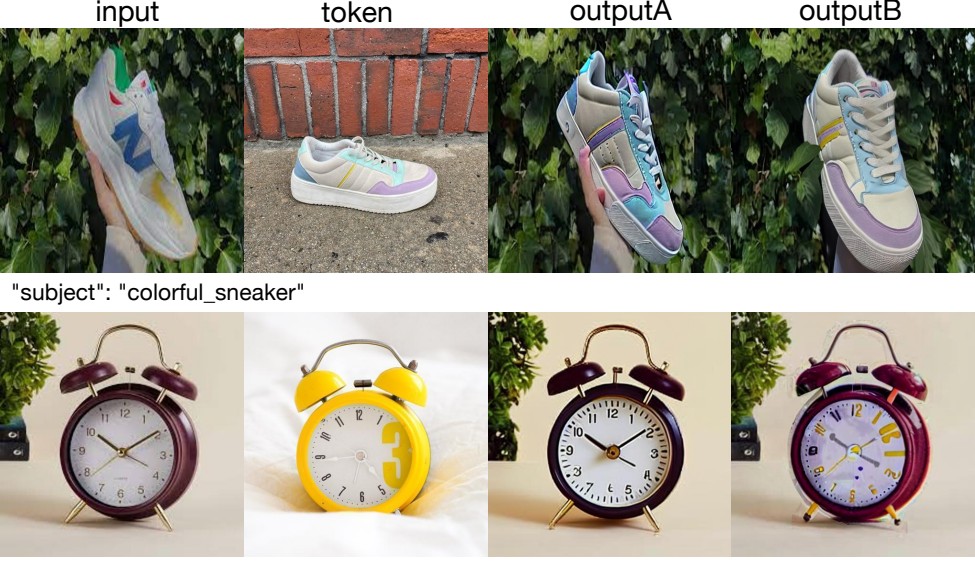

Figure 10: Rating examples on Subject-driven Image Editing task.

- Output$A_1$: [0.5, 0.5]. SC=0.5: Some details of the subject do not match the token. PR=0.5: Some unnatural sense on the hand features.

- Output$B_1$: [0.5, 1]. SC=0.5: The subject does match the token, but removing the hand is an unnecessary edit. PR=1: It looks real.

- Output$A_2$: [0, 0.5]. SC=0: Subject clock does not match. PR=0.5: Some distortion on the numbers of the clock.

- Output$B_2$: [0, 0.5]. SC=0: Subject clock does not match. PR=0.5: Some distortion on the numbers of the clock and the edit region does not blend well with the context.

**Control-guided Image Generation.**

input        outputA        outputB

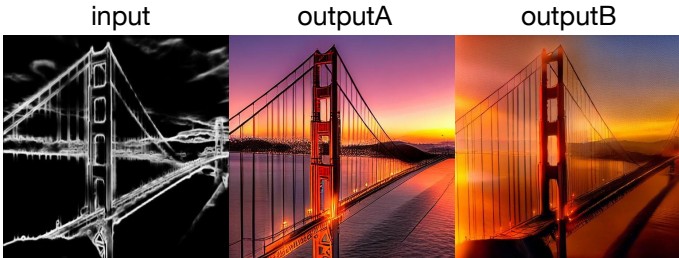

"prompt": "golden gate bridge at sunset, Golden Gate Bridge in San Francisco, USA",
"control_type": "hed".

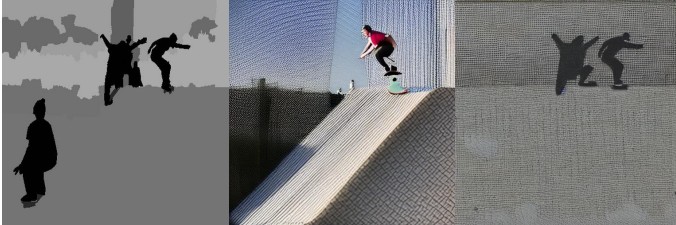

"text": "a man riding a skateboard up the side of a ramp",
"control_type": "depth".

Figure 11: Rating examples on Control-guided Image Generation task.

- OutputA$_1$: [1, 0.5]. SC=1: The generated image perfectly described all the required attributes of the user prompt, and even work. PR=0.5: There are some missing details in the background. The details in the far-side are mostly blurred unnaturally.

- OutputB$_1$: [1, 0.5]. SC=1: The generated image perfectly described all the required attributes of the user prompt. PR=0.5: The details of golden gate bridge are mostly blurred unnaturally.

- OutputA$_2$: [0, 0]. SC=0: The output does not correspond to the controlled depth image. PR=0: Heavy distortion on the background and the man.

- OutputB$_2$: [0, 0]. SC=0: The meaning of the text cannot be obtained from the output. PR=0: Heavy distortion.

## A.4 DATASET INFORMATION

**Dataset Details.**

**Dataset Distribution.**

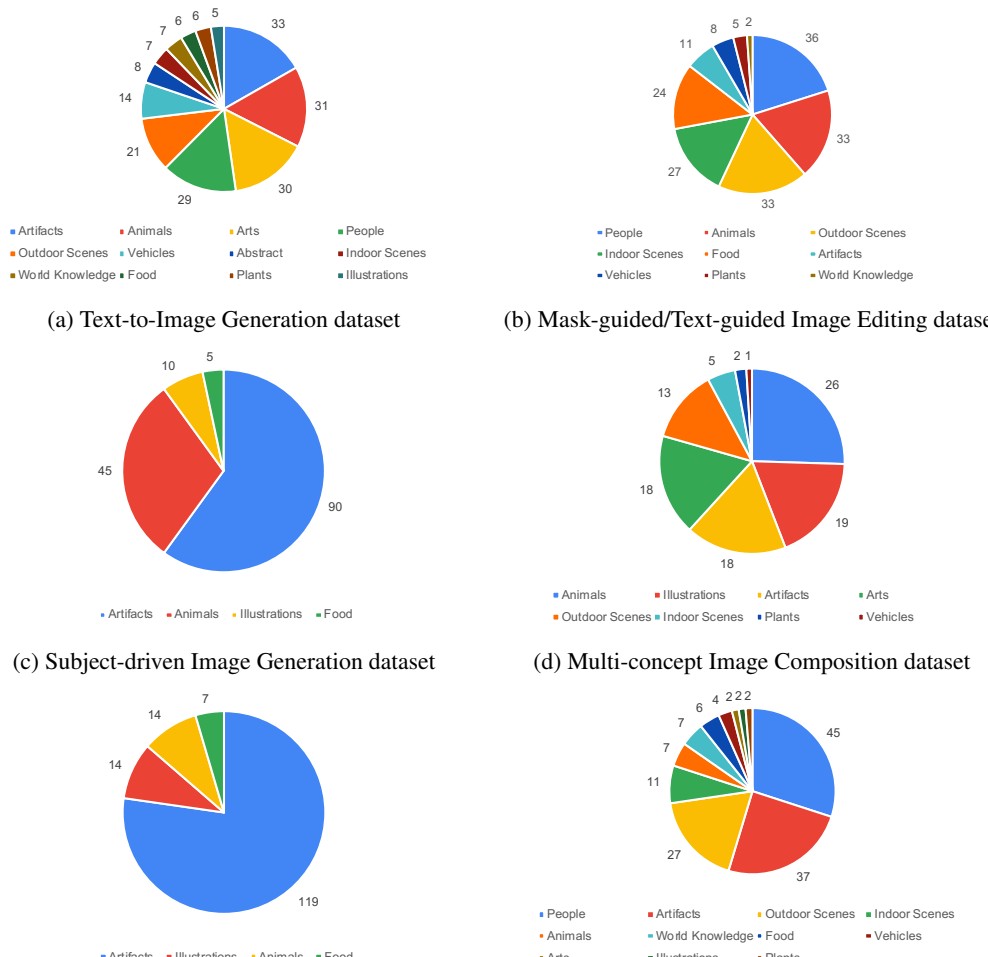

(a) Text-to-Image Generation dataset

(b) Mask-guided/Text-guided Image Editing dataset

(c) Subject-driven Image Generation dataset

(d) Multi-concept Image Composition dataset

(e) Subject-driven Image Editing dataset

(f) Control-guided Image Generation dataset

Figure 12: Objects presented in each task's prompt text.

## A.5 VISUALIZATION

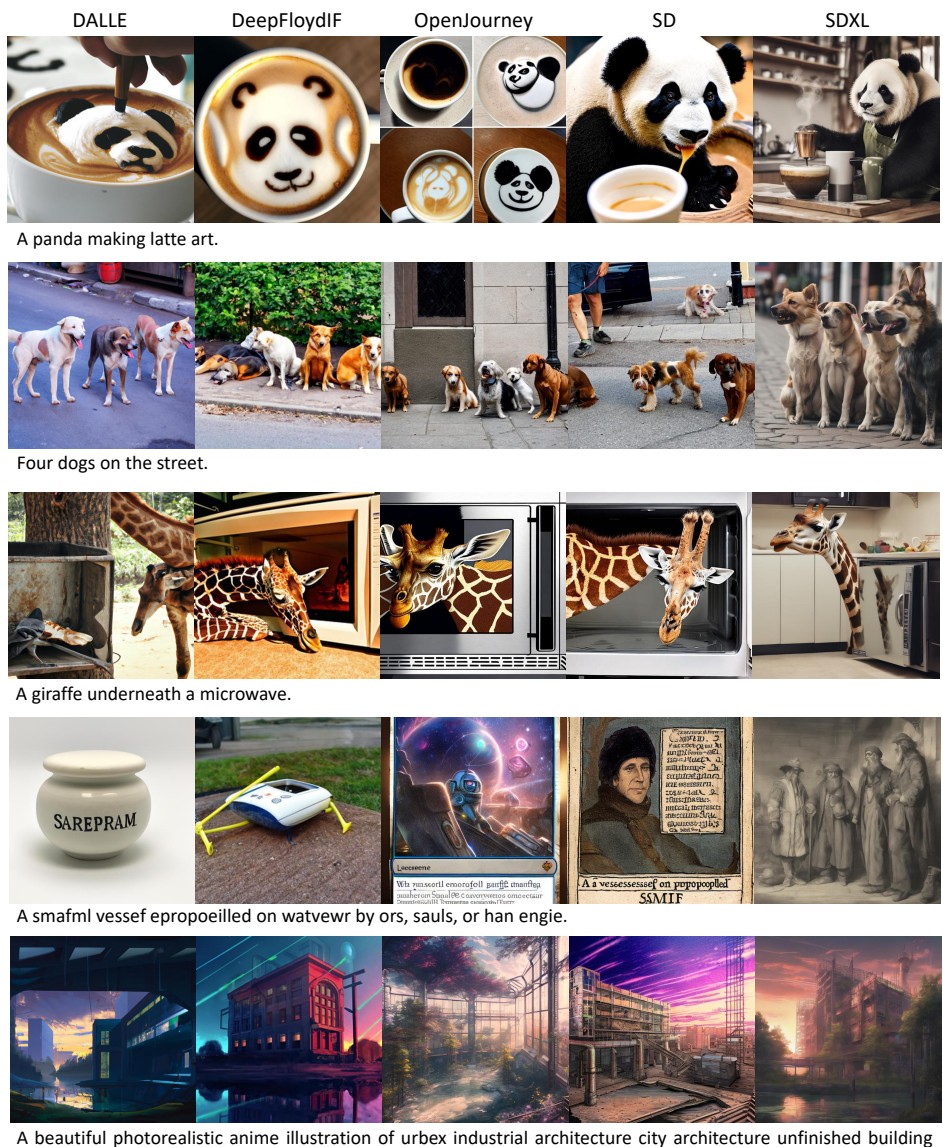

Figure 13: More samples from our Text-guided Image Generation dataset and ImagenHub outputs.

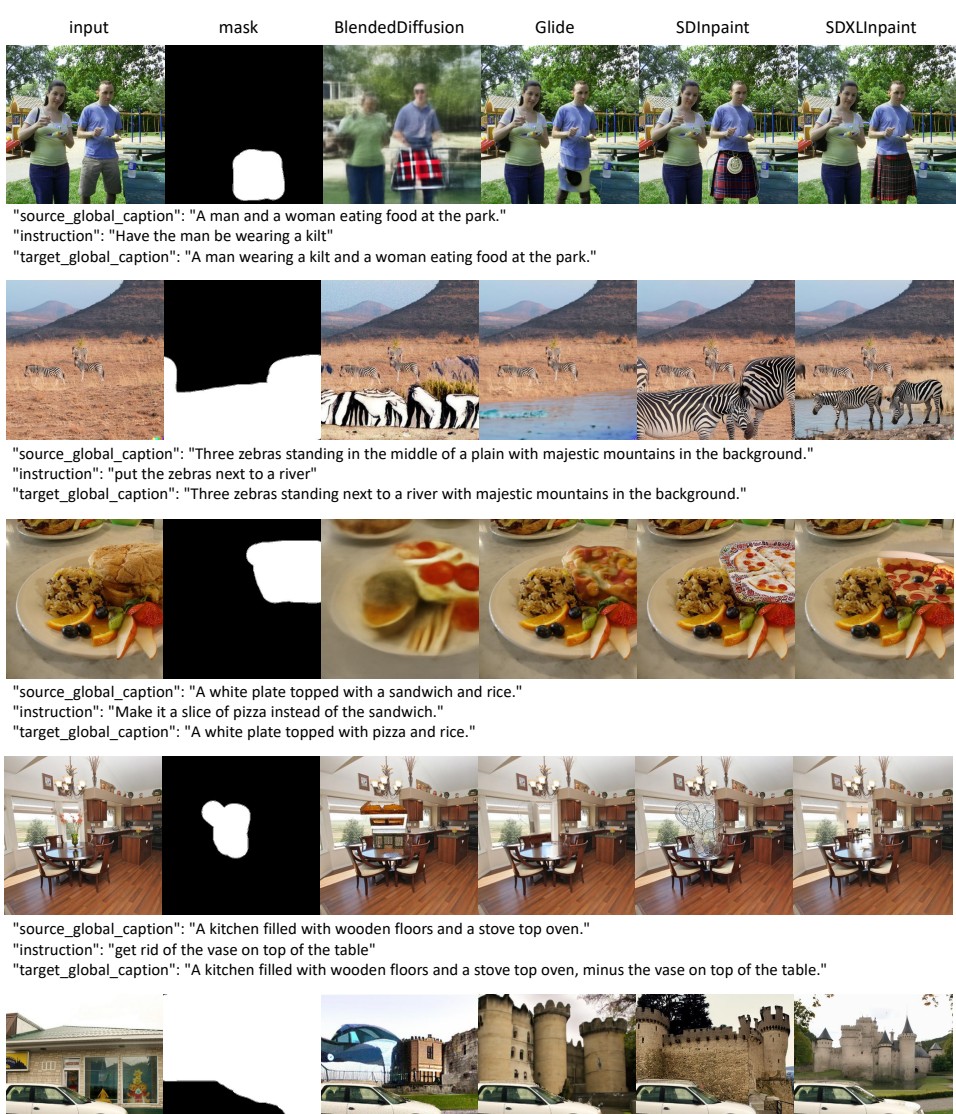

input    mask    BlendedDiffusion    Glide    SDInpaint    SDXLInpaint

"source_global_caption": "A man and a woman eating food at the park."
"instruction": "Have the man be wearing a kilt"
"target_global_caption": "A man wearing a kilt and a woman eating food at the park."

"source_global_caption": "Three zebras standing in the middle of a plain with majestic mountains in the background."
"instruction": "put the zebras next to a river"
"target_global_caption": "Three zebras standing next to a river with majestic mountains in the background."

"source_global_caption": "A white plate topped with a sandwich and rice."
"instruction": "Make it a slice of pizza instead of the sandwich."
"target_global_caption": "A white plate topped with pizza and rice."

"source_global_caption": "A kitchen filled with wooden floors and a stove top oven."
"instruction": "get rid of the vase on top of the table"
"target_global_caption": "A kitchen filled with wooden floors and a stove top oven, minus the vase on top of the table."

"source_global_caption": "Car parked in parking lot in front of a building."
"instruction": "edit the background by removing the museum and placing a castle"
"target_global_caption": "Car parked in front of a castle in a parking lot."

Figure 14: More samples from our Mask-guided Image Editing dataset and ImagenHub outputs.

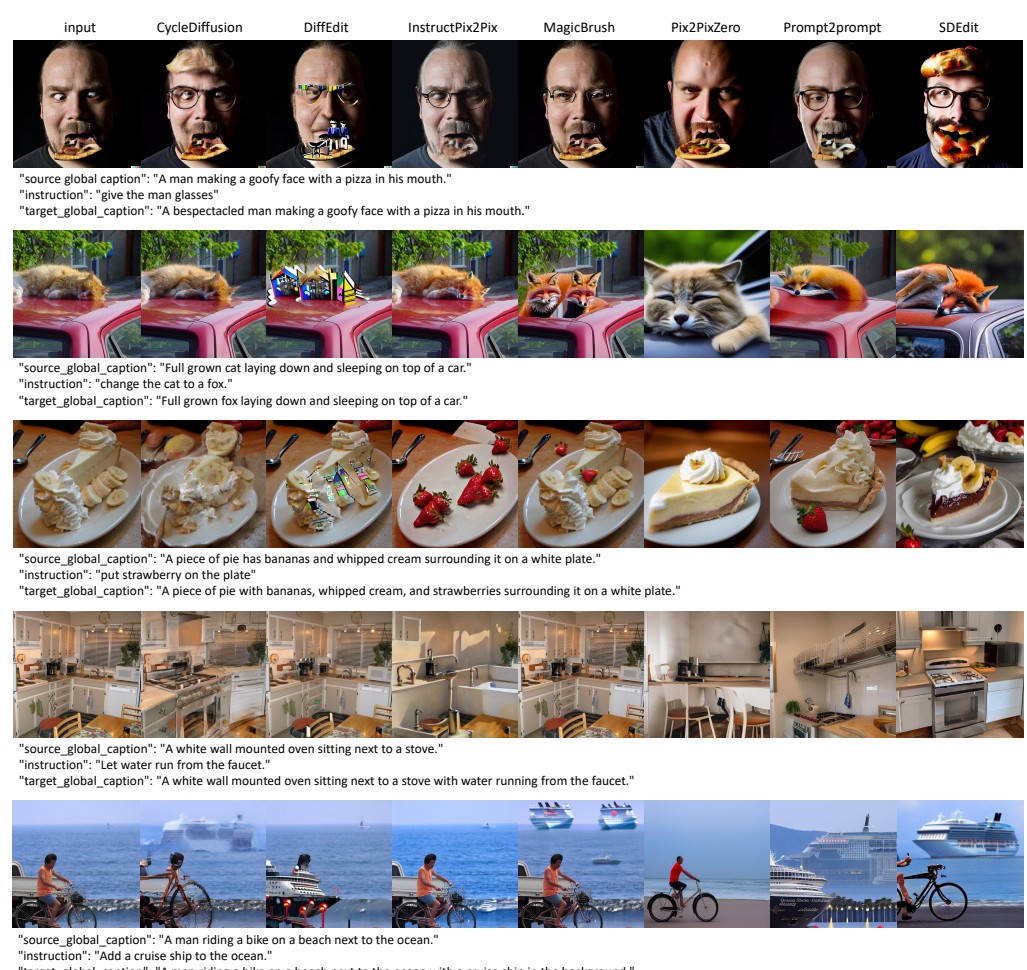

input    CycleDiffusion    DiffEdit    InstructPix2Pix    MagicBrush    Pix2PixZero    Prompt2prompt    SDEdit

"source global caption": "A man making a goofy face with a pizza in his mouth."
"instruction": "give the man glasses"
"target_global_caption": "A bespectacled man making a goofy face with a pizza in his mouth."

"source_global_caption": "Full grown cat laying down and sleeping on top of a car."
"instruction": "change the cat to a fox."
"target_global_caption": "Full grown fox laying down and sleeping on top of a car."

"source_global_caption": "A piece of pie has bananas and whipped cream surrounding it on a white plate."
"instruction": "put strawberry on the plate"
"target_global_caption": "A piece of pie with bananas, whipped cream, and strawberries surrounding it on a white plate."

"source_global_caption": "A white wall mounted oven sitting next to a stove."
"instruction": "Let water run from the faucet."
"target_global_caption": "A white wall mounted oven sitting next to a stove with water running from the faucet."

"source_global_caption": "A man riding a bike on a beach next to the ocean."
"instruction": "Add a cruise ship to the ocean."
"target_global_caption": "A man riding a bike on a beach next to the ocean with a cruise ship in the background."

Figure 15: More samples from our Text-guided Image Editing dataset and ImagenHub outputs.

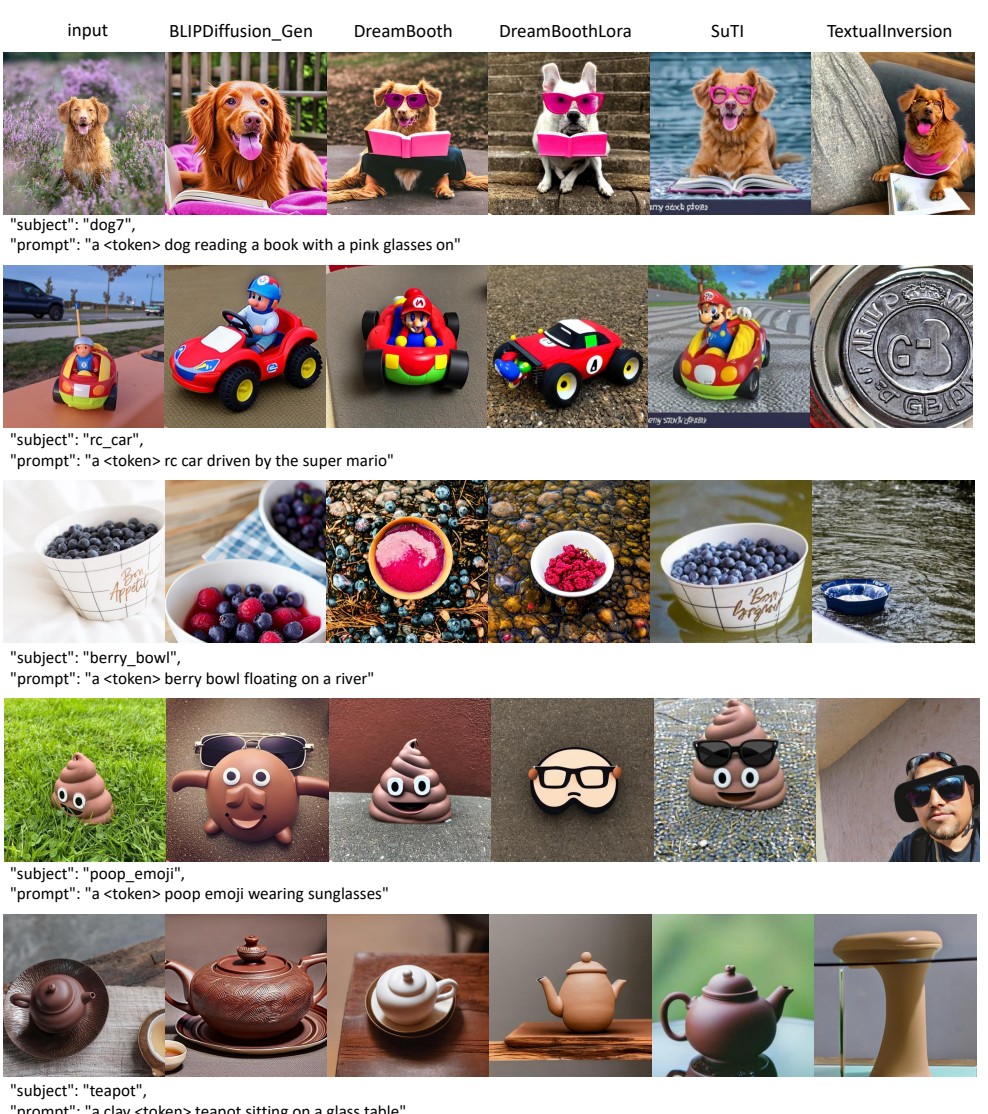

|  | input | BLIPDiffusion_Gen | DreamBooth | DreamBoothLora | SuTI | TextualInversion |

"subject": "dog7",
"prompt": "a <token> dog reading a book with a pink glasses on"

"subject": "rc_car",
"prompt": "a <token> rc car driven by the super mario"

"subject": "berry_bowl",
"prompt": "a <token> berry bowl floating on a river"

"subject": "poop_emoji",
"prompt": "a <token> poop emoji wearing sunglasses"

"subject": "teapot",
"prompt": "a clay <token> teapot sitting on a glass table"

Figure 16: More samples from our Subject-Driven Image Generation dataset and ImagenHub outputs.

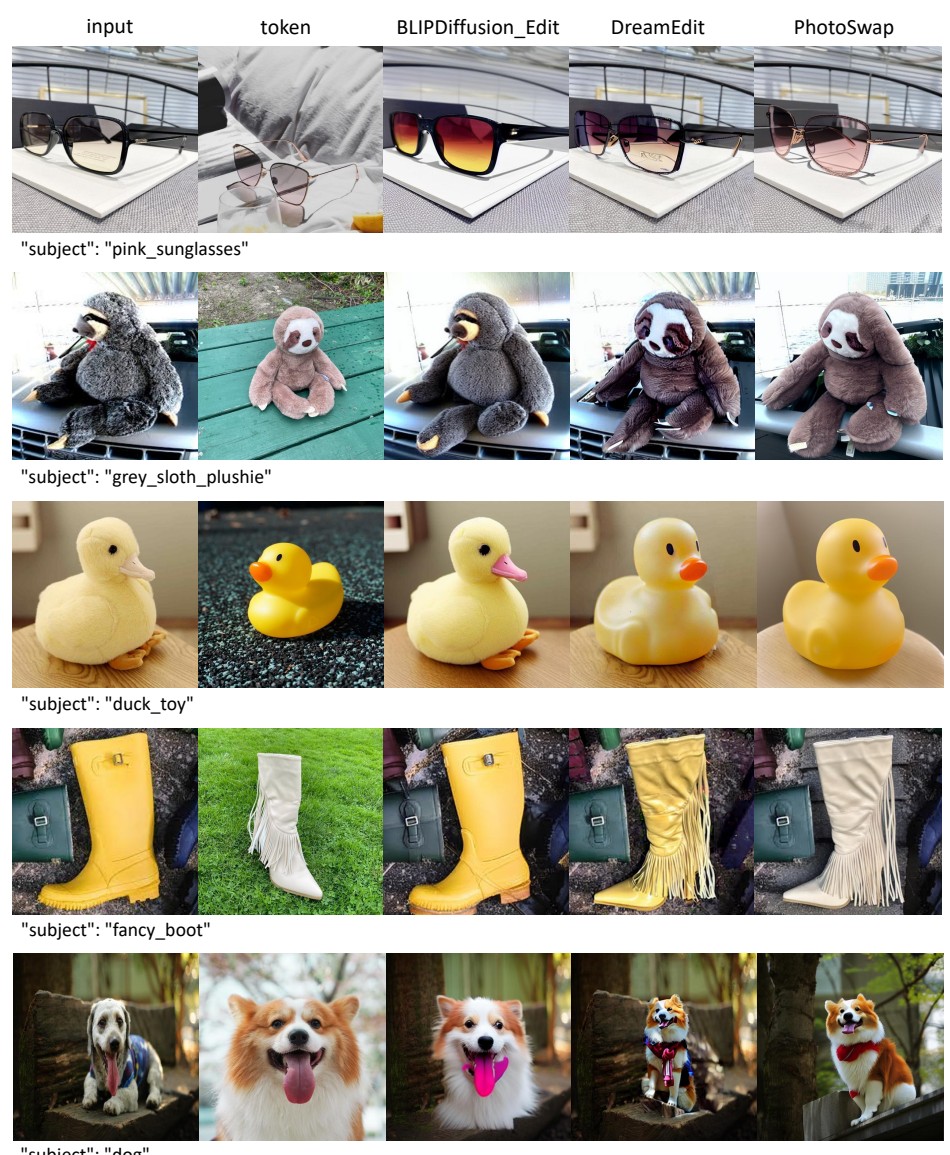

Figure 17: More samples from our Subject-Driven Image Editing dataset and ImagenHub outputs.

| input | CustomDiffusion | DreamBooth | TextualInversion |
|---|---|---|---|

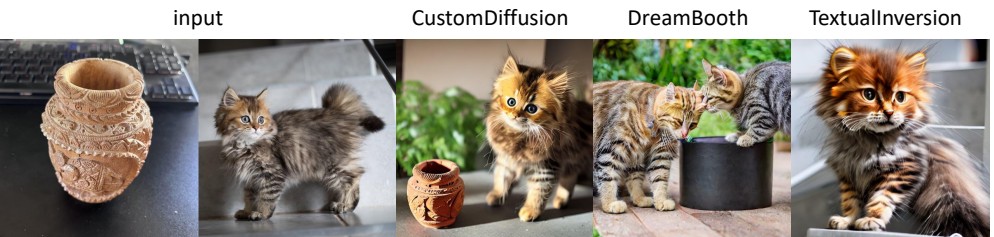

"prompt": "the cat playing with a wooden pot in a garden",
"concept1": "wooden pot",
"concept2": "cat"

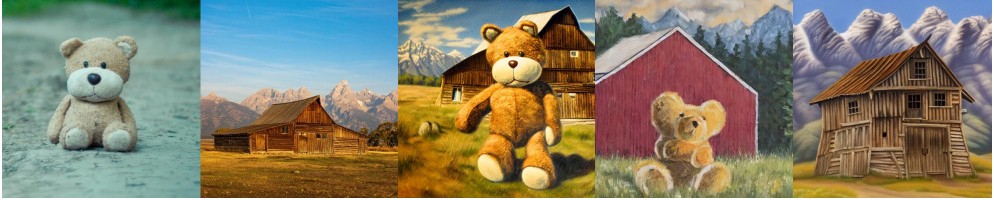

"prompt": "Oil painting of a teddybear in front of the barn",
"concept1": "teddybear",
"concept2": "barn"

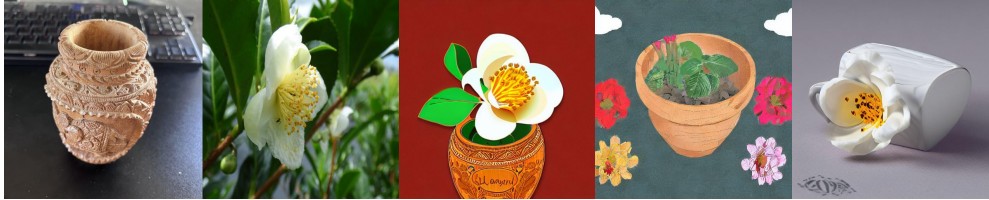

"prompt": "a digital illustration of flower in wooden pot",
"concept1": "wooden pot",
"concept2": "flower"

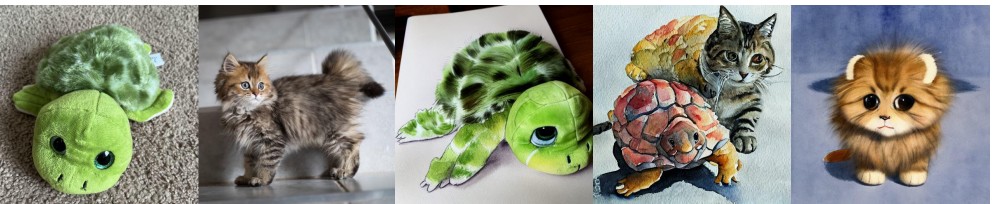

"prompt": "Watercolor painting of tortoise plushy next to a cat",
"concept1": "tortoise plushy",
"concept2": "cat"

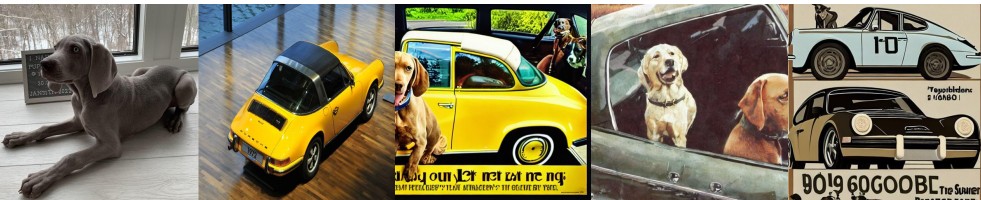

"prompt": "Vintage poster of a car with a dog in the backseat",
"concept1": "dog",
"concept2": "car"

Figure 18: More samples from our Multi-Concept Image Composition dataset and ImagenHub outputs.

| input | ControlNet | UniControl |
|-------|------------|------------|

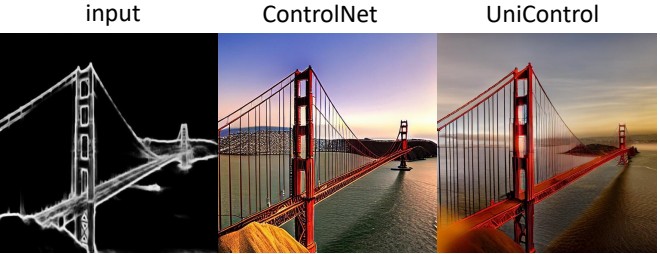

Golden gate bridge san francisco, Golden Gate Bridge in San Francisco, USA

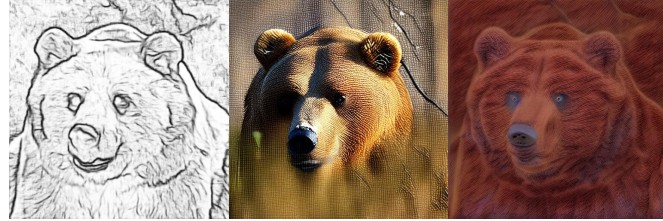

Closeup of a brown bear sitting in a grassy area.

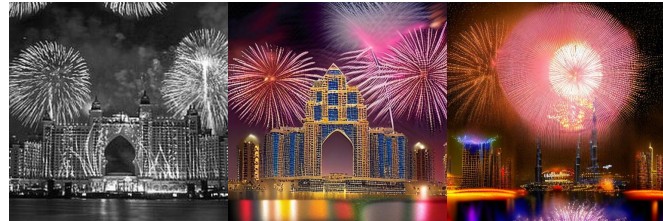

"New Year's Eve: Dubai Will Attempt ""Largest Fireworks Display"" World Record"

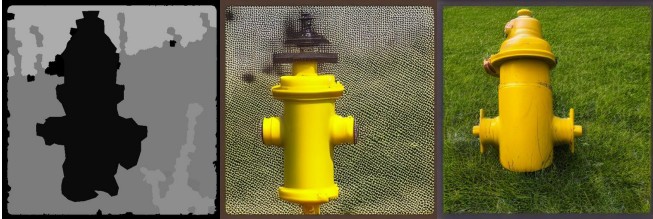

A yellow fire hydrant sitting in the grass

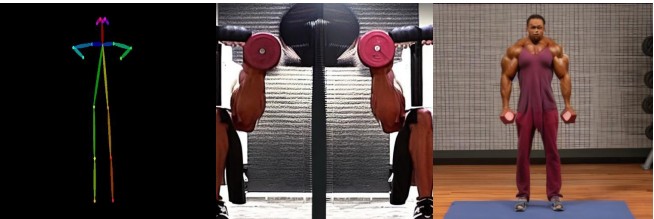

A man standing on a mat with a pair of dumbbells, deep fake, youtube video screenshot, off the shoulder shirt, underbrush wash, two legs two arms one head, neck shackle, grainy footage, surfaces blemishes, artem, uses c4, buttshape, sconces, threes, head to waist

Figure 19: More samples from our Control-guided Image Generation dataset and ImagenHub outputs.

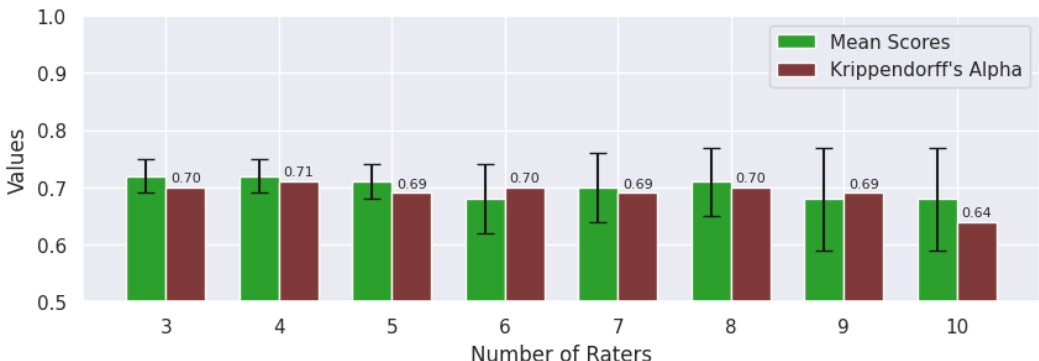

Figure 20: Mean scores of overall human rating and Krippendorff's Alpha for each number of raters on one model.

## A.6 HUMAN EVALUATION PROCEDURE

**Prolific.** In this work, we used the Prolific platform to receive human feedback on rating AI-generated images. We applied the following filter for worker requirements when creating the project: 1) The Worker's primary speaking language has to be English. 2) Worker must complete rating one set of images (around 150-200 images) in one go. We do that to ensure workers can fully understand the rating instructions, and each set of image is rated by the same person so we can perform Inter-rater Reliability Analysis across the whole set of the benchmark. Workers will rate each image in [SC, PQ], while SC in [0, 0.5, 1] and PQ in [0, 0.5, 1]. Instructions and images are hosted on a website. Workers are provided with detailed instructions and rating examples to assist their rating process. Workers will submit the responses in a tsv file. To validate the annotation quality, we selected a few (around 10-20) significant samples with obvious ratings for each task. We rated them by ourselves and then used them as a reference to determine the performance of annotators. Poor performance will be rejected.

**Sample Size Consideration of Human Raters.**

We studied the effect on the number of human raters. We recruited 10 human raters on Prolific to rate 15 samples generated using the Midjourney model. In Figure 20, the mean scores seem to remain relatively constant across the number of raters, with a slight increase at 4 raters and the highest mean score at 5 raters. The values for the mean scores are hovering around 0.70, as the error bar shows the standard deviation increases. On the other hand, Krippendorff's Alpha generally decreases as the number of raters increases. It starts at around 0.70 with 3 raters and has a noticeable drop at 10 raters to 0.64. We can observe that the mean score will be always consistent, while the reliability of that score as measured by Krippendorff's Alpha decreases slightly as more raters are involved, as more variance is introduced. To make our human evaluation protocol efficient, we picked 3 as the number of human raters.

## A.7 VISUALIZATION TOOLS (IMAGEN MUSEUM)

**Experiment Transparency.** We are hosting the benchmarking results publicly. Thus showing the true performance of each model. Viewers can gain their insights from the result by looking at the actual results. Our framework encourages future researchers to release the human evaluation set generation results publicly. By standardizing the human evaluation protocol, human evaluation results would become far more convincing with the experiment transparency.

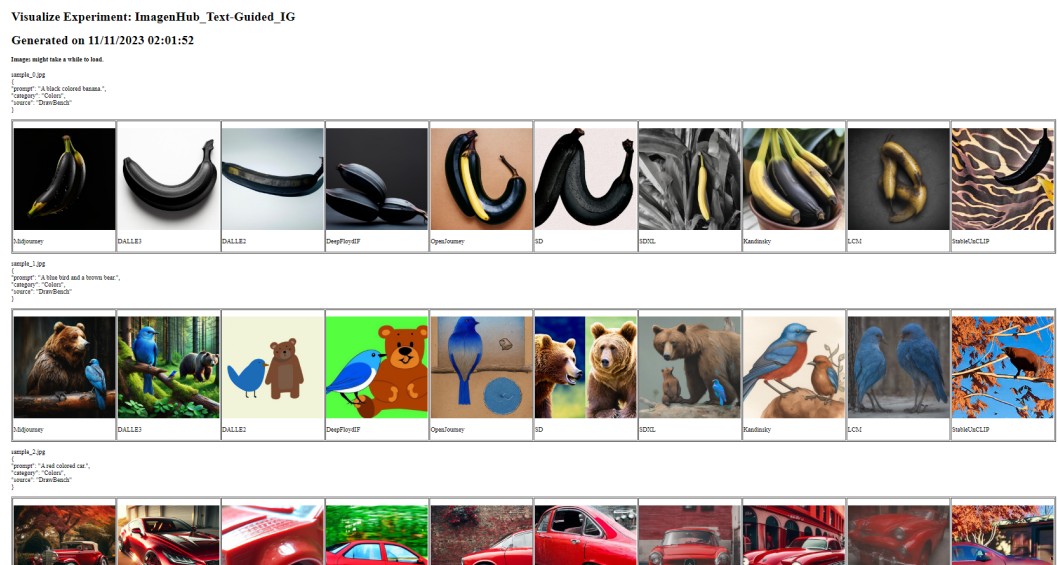

Figure 21: The webpage design of Imagen Museum (Text-Guided Image Generation page).

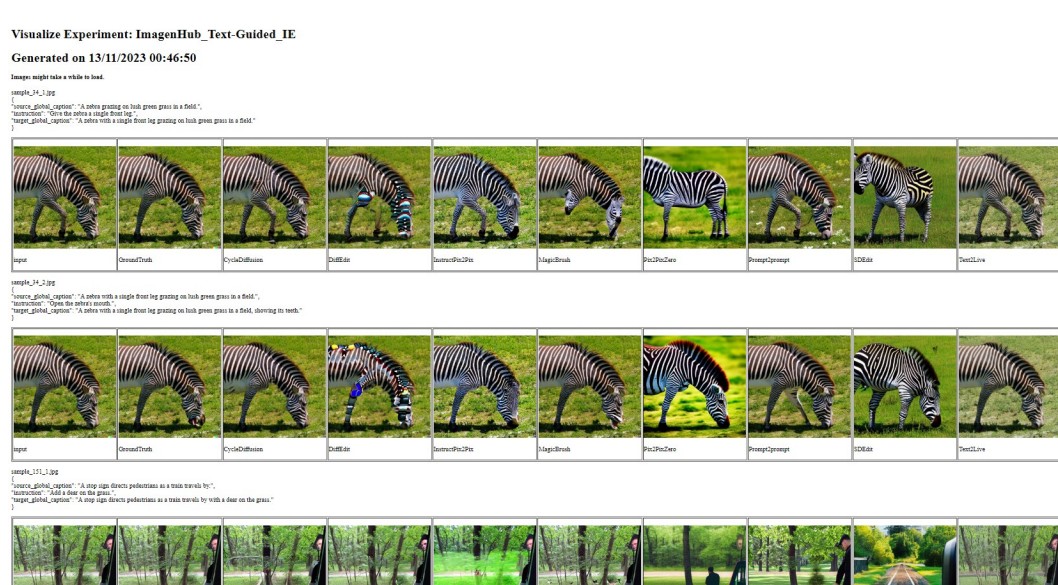

Figure 22: The webpage design of Imagen Museum (Text-Guided Image Editing page).

Please refer to https://github.com/ChromAIca/ChromAIca.github.io for the visualization results.

## A.8 ADDING NEW MODELS TO OUR FRAMEWORK.

Third parties are welcome to use our provided tools and frameworks for their work. Here is a quick walkthrough of how a third-party member can extend the framework to support their own model's inference. We use the Text-Guided Image Generation (Text-To-Image) task as an example for the sake of simplicity.

The `infermodel` class is designed to have the following two methods:

- `__init__(args)` for class initialization.
- `infer_one_image(args)` to produce 1 image output. Please try to set the seed as 42.

In that case, you will add a new file in `imagen_hub/infermodels` folder. `imagen_hub/infermodels/awesome_model.py`

Then a line can be added in `imagen_hub/infermodels/__init__.py`:

And modify the template config.yml file and add your own model.

Finally run

Please refer to https://imagenhub.readthedocs.io/en/latest/Guidelines/custommodel.html for details.

## A.9 IMAGENHUB AS AN INFERENCE LIBRARY.

**Custom inference of multiple models.**

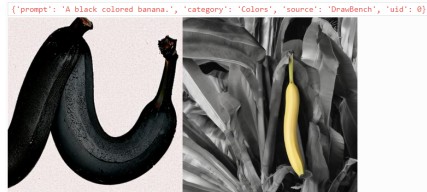

Figure 23: Image generated in this custom inference example.

**Evaluate images with autometrics.**

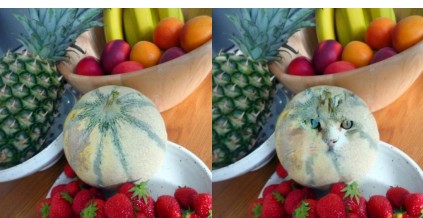

Figure 24: Image evaluated in this autometrics example.

Please refer to https://imagenhub.readthedocs.io/en/latest/Guidelines/deepdive.html for details.

