# OpenReview forum: "ImagenHub: Standardizing the evaluation of conditional image generation models"
_ICLR.cc/2024/Conference — ICLR 2024 poster_

### Official Review · Reviewer_teCT · 2023-10-26

**Soundness:** 3 good
**Presentation:** 3 good
**Contribution:** 3 good
**Rating:** 6
**Confidence:** 3

**Summary:**

Paper proposes a library for the evaluation of coditional image generation models. They consider 7 tasks (text-guided, subject-driven, control-gruided etc). They define two human evaluation scores (semantic consistency and perceptual quality), train the raters, and evaluate around 30 models for the various tasks.

**Strengths:**

1. paper is well written and presentation is good.
2. a fair comparison based on human raters is interesting for many users and scientists in this dense research field.
3. evaluation setup and comparisons are well-designed and seem fair (many methods use their own curated datasets and are here compared on the same data).

**Weaknesses:**

1. I think the paper would have been stronger if the authors would have directly compared the human evaluations with existing automatic evaluation metrics. It would be very interesting to know the correlations.

2. Ideally we would have computable metrics which correlate high with human evaluations. The paper does not explain very well what the main problems are of existing metrics. It would also be interesting to see what parts of human evaluation are missed by currently used metrics.

3. It remains unclear how a third party with a new method could make use of this benchmark since it is based on human raters. It would be nice if there is some rater-training guide which would allow other researchers to also evaluate their method on the proposed benchmark. There are no safeguards for the maintenance of the benchmark by the authors.

4. I found the discoveries and insights not especially surprising. They were often more based on looking at results than referring to the human evaluation rates.

**Questions:**

I think the study is of interest for many people. However, I found the technical contribution still a bit shallow and the possible usage for future model evaluation unclear. If some results on weakness point 1 could be added to the paper, I would probably be willing to raise my score.

- please address the mentioned weaknesses.

minor remarks:
- ref to Table 10 in section 4.2 is wrong (should be table 2)
- spellcheck 'Inference: ' text on page 2.

-------------------------------
POST REBUTTAL:
I thank the reviewers for their feedback. I apologize for mising Table 8 in Appendix. And I appreciate the effort done to ensure ongoing usefulness of the proposed benchmark for future users (A7-A9). I have raised my score.

---

> ### Author Response · Authors · 2023-11-17
> **Response to Reviewer teCT**
>
> (w1) I think the paper would have been stronger if the authors had directly compared the human evaluations with existing automatic evaluation metrics. It would be very interesting to know the correlations...
>
> We have compared the correlations in Table 8: Metrics correlation. (Appendix A1)
>
> (w2) Ideally we would have computable metrics which correlate highly with human evaluations. The paper does not explain very well what the main problems are of existing metrics. It would also be interesting to see what parts of human evaluation are missed by currently used metrics...
>
> The computable metrics all hold different assumptions which only contribute part of the aspects in human evaluation. This makes a low correlation with human evaluations. For example, CLIPscore only measures the compatibility of image-caption pairs. But there are more aspects from the human perspective of semantic consistency. In editing tasks there are the level of editing, in subject-driven tasks it has to measure whether the subject aligns. Those are still an open problem in the field. Thus we found a low correlation for CLIP score across all tasks in Table 8. On the other hand, perceptual quality metrics like FID focused on visual quality and LPIPS evaluates the distance between image patches. While they are good at detecting distortion and artifacts, From the human perspective of perceptual quality, the overall natural sense of whole image is often considered.
>
> (w3) It remains unclear how a third party with a new method could make use of this benchmark since it is based on human raters. It would be nice if there is some rater-training guide which would allow other researchers to also evaluate their method on the proposed benchmark. There are no safeguards for the maintenance of the benchmark by the authors...
>
> There are two ways to evaluate their methods on ImagenHub. The preferred option is that the third party will send a pull request to our GitHub repo to add their model inference interface. We will use their code to perform standaridized evaluation and then publish our leaderboard.
> The less preferred option is that the third party will follow our human judgment guide and hire their own raters to do it. Our guidelines and examples are in Appendix A2 and A3. We also added Appendix A7, A8, and A9 to show how a third party with a new method could make use of the benchmark. Links are hidden due to the Anonymous policy.
>
>
> (w4) I found the discoveries and insights not especially surprising. They were often more based on looking at results than referring to the human evaluation rates...
>
> We think there are some interesting insights: (1) our rigorous eval shows that the existing methods despite their popularity are actually doing very poorly on several generation tasks. There is still a large room for improvement, (2) we went through the claims in previous papers and found that 17% of them are ill-posed due to rigorous human evaluation, (3) we found that besides T2I and subject-driven generation, the automatic metrics are having extremely low correlation with human raters. It indicates that we need to develop better metrics for those tasks.
>
> (q1) I think the study is of interest for many people. However, I found the technical contribution still a bit shallow and the possible usage for future model evaluation unclear. If some results on weakness point 1 could be added to the paper, I would probably be willing to raise my score...
>
> Our main technical contribution focused on the inference library and the unified inference and evaluation protocols. Although we have already released the codebase on GitHub, we cannot mention the links due to the Anonymous policy. In this revised version, we added Appendix A7, A8, and A9 to show how a third party and researchers can benefit from our work and easily extend it.

---

> > ### Author Response · Authors · 2023-11-21
> > **Follow-up**
> >
> > Dear Reviewer teCT,
> > I am writing to inquire if you have had the chance to review our revised manuscript and the response to your comments. We greatly value your feedback and would appreciate any further questions or comments you might have.
> > Sincerely,

---

> ### Comment · Reviewer_teCT · 2023-11-21
> **rebuttal**
>
> I thank the reviewers for their feedback. I apologize for mising Table 8 in Appendix. And I appreciate the effort done to ensure ongoing usefulness of the proposed benchmark for future users (A7-A9). I have raised my score.

---

### Official Review · Reviewer_JKzK · 2023-10-28

**Soundness:** 3 good
**Presentation:** 3 good
**Contribution:** 3 good
**Rating:** 8
**Confidence:** 3

**Summary:**

This paper proposes ImagenHub, a new benchmark for conditional image generation based on human evaluators.
The benchmark evaluates seven conditional image generation tasks.
During evaluation, human evaluators will follow two major metrics, namely semantic consistency and perceptive quality.
The formal metric (SC) ensures that the generated image is aligned with the given condition, while the second metric (PQ) ensures that the generated image is of good visual quality.
The authors evaluate major opensource image generation approaches based on the two metrics and report the results.

**Strengths:**

1. This paper first proposes a comprehensive benchmark to evaluate different conditional image generation tasks. It evaluates a bunch of image generation models with human evaluators, and provides comprehensive evaluation results to the community.

**Weaknesses:**

1. It would be better if the authors can involve more top-performing image generation approaches in this comparisons (though some of them may not be opensource), like MidJourney or DALLE-3 (in a later revision of the paper).
2. Some previous works (like T2I CompBench) have also proposed some evaluation metrics for benchmarking conditional image generation models (related to the Semantic Consistency part in this paper). It would be better to discuss and see if such metrics result in similar trends compared to the human-based evaluation.

**Questions:**

1. How do the authors validate the annotation quality, and the measure the performance of annotators?
2. How much human labor is required to build this benchmark (in annotator x hour)?

---

> ### Author Response · Authors · 2023-11-17
> **Response to Reviewer JKzK**
>
> (w1) It would be better if the authors can involve more top-performing image generation approaches in this comparisons (though some of them may not be opensource), like MidJourney or DALLE-3 (in a later revision of the paper).
>
> We have gathered and evaluated the generation results of MidJourney and DALLE-3 on our ImagenHub benchmark. Please refer to B1 in our new paper version. It turns out that DALLE-3 and Mijourney are extremely good, outperforming the other methods by an absolute overall score of 10-20%.
>
> (w2) Some previous works (like T2I CompBench) have also proposed some evaluation metrics for benchmarking conditional image generation models (related to the Semantic Consistency part in this paper). It would be better to discuss and see if such metrics result in similar trends compared to the human-based evaluation.
>
> T2I CompBench’s proposed metric is specifically for Text-To-Image task. They cannot be used across all the ImagenHub defined tasks. For example, UniDet (from T2I CompBench) does not work on Subject-Driven Image Editing and Generation tasks. Our paper focused on evaluation metrics that apply to most of the tasks.
>
> (q1) How do the authors validate the annotation quality, and the measure the performance of annotators?
>
> We selected a few (around 10-20) significant samples with obvious ratings. We rated them by ourselves and then used them as a reference to determine the performance of annotators.
>
> (q2) How much human labor is required to build this benchmark (in annotator x hour)?
>
> * The current benchmark (total of 30 models) was built by 24 annotators in a total of an estimated 150 hours. Each of the annotators conducted evaluations on 3-4 sets (or more) of images.
> * We required an annotator to rate one set of images in one go.
> * For the time used in evaluating one set of images (around 150-200 images). The mean time reported is 1 hour 40 minutes.

---

> > ### Author Response · Authors · 2023-11-21
> > **Follow-up**
> >
> > Dear Reviewer JKzK, I am writing to inquire if you have had the chance to review our revised manuscript and the response to your comments. We greatly value your feedback and would appreciate any further questions or comments you might have. Sincerely,

---

> > > ### Comment · Reviewer_JKzK · 2023-11-22
> > > **Rebuttal**
> > >
> > > Dear authors,
> > > I appreciate the additional results of Midjourney and Dalle-3 you provided in the rebuttal. I will raise my score accordingly.

---

### Official Review · Reviewer_gWLk · 2023-10-31

**Soundness:** 4 excellent
**Presentation:** 4 excellent
**Contribution:** 4 excellent
**Rating:** 8
**Confidence:** 4

**Summary:**

This paper presents ImagenHub, a dataset and library for standardized evaluation of conditional image generation models. A large amount of models is evaluated using a unified evaluatoin protocol of human raters. Two metrics are proposed to judge semantic consistency and perceptual quality, and the evaluation protocol is adjusted for high inter-worker agreement.

**Strengths:**

- The paper tackles an important problem, namely inconsistent evaluation protocols of the large amount of recent image generation methods.
- The paper presents a sound approach for fair comparison using human raters.
- The paper contributes a library to standardize and ease the evaluation of future generative models.

**Weaknesses:**

- The paper states that 83% of the published results are consistent with the ranking, and the presented evaluation results often validate the results of published works.
  - a) Where does this 83% come from?
  - b) What about the other 17%? What kind of results from published work is not consistent with the presented work? Is it due to limitations of the presented paper or wrong claims by published work?

Two minor points:
- It would be beneficial to incorporate more automatic measure such as the commonly used FID or detection based scores to evaluate spatial fidelity, object recognizability as well as counts of objects.
- It could be interesting to see an analysis on the costs and time needed of such a unified evaluation protocol given that the method relies on human raters.

**Questions:**

- Is it possible to analyze drift of user ratings over time? In other words, how much is the rating influenced by the experience/exposure of a rater to the evaluation platform?

---

> ### Author Response · Authors · 2023-11-17
> **Response to Reviewer gWLk**
>
> (w1a) The paper states that 83% of the published results are consistent with the ranking ... Where does this 83% come from?
>
> We validated the claims in Section 5.1. We compared the rankings of human evaluation and automatic rankings to the published works. 25 models aligned with our results thus 25/30 ~= 83%.
>
> (w1b) ... What about the other 17%? What kind of results from published work is not consistent with the presented work? Is it due to limitations of the presented paper or wrong claims by published work?
>
> We mentioned the works that were not consistent with the presented work in Section 5.1. We compared the rankings of human evaluation and automatic rankings to the published works. There is a total of 5/30 ~= 17%. The 5 of them are Pix2PixZero, DreamEdit, UniControl, DiffEdit, and BLIP-Diffusion. The inconsistency is mainly due to their human evaluation not being conducted rigorously enough. For example, some of the papers only use single raters for each instance and do not inter-rate agreement. Some of the papers heavily tune the hyper-parameters on the instance level, which is not allowed in our setting where we only allow fixed hyper-parameters.
>
> (w2) It would be beneficial to incorporate more automatic measure such as the commonly used FID or detection based scores to evaluate spatial fidelity, object recognizability as well as counts of objects.
>
> We did not have FID score because FID score requires a large set of reference images to compute. For most tasks, our reference set is very small and the FID score might not be statistically significant enough.
> Per the request, we have added KID and FID scores in our revised paper (Appendix A1). Our current benchmark does not have enough data for detection-based scores.
>
> (w3) It could be interesting to see an analysis of the costs and time needed for such a unified evaluation protocol given that the method relies on human raters.
>
> * For the time used in evaluating one set of images (around 150-200 images). The mean time reported is 1 hour 40 minutes.
> * We paid $25 (£20) for each set of images.
>
> (q1) Is it possible to analyze drift of user ratings over time? In other words, how much is the rating influenced by the experience/exposure of a rater to the evaluation platform?
>
> That is an interesting problem. We believe that tracking the ratings of each rater across different time points might help analyze the drift of user ratings. From our observation, the user rating is rather often affected by the leniency, and the leniency can be driven by multiple factors such as screen resolution, the rater’s eyesight condition, and mood etc.. However, we also observed that does not create a large impact on the overall mean rating due to our multiple raters policy.

---

> > ### Author Response · Authors · 2023-11-21
> > **Follow-up**
> >
> > Dear Reviewer gWLk,
> > I am writing to inquire if you have had the chance to review our revised manuscript and the response to your comments. We greatly value your feedback and would appreciate any further questions or comments you might have.
> > Sincerely,

---

> > > ### Comment · Reviewer_gWLk · 2023-11-22
> > > **Response to authors**
> > >
> > > Dear authors, I appreciate the reply and provided information to answer my questions. I have raised my confidence.

---

### Official Review · Reviewer_nZYf · 2023-11-05

**Soundness:** 2 fair
**Presentation:** 3 good
**Contribution:** 2 fair
**Rating:** 5
**Confidence:** 3

**Summary:**

The paper introduces ImagenHub, a standardized framework for evaluating conditional image generation and editing models, addressing inconsistencies in experimental conditions. It defines key tasks, creates evaluation datasets, establishes a unified inference pipeline, and introduces human evaluation scores. Results indicate that existing models generally perform poorly, except for Text-guided and Subject-driven Image Generation, and validate most claims from published papers while highlighting the inadequacy of existing automatic metrics, with plans to continue evaluating new models and tracking progress in the field.

**Strengths:**

This is an extensive endeavor! Comparison of several models plus human evaluation is presented. It is an important problem and the this is a timely study. In general, I am leaning towards accepting the paper but there are several issues and questions that need to be addressed. I would like to see the authors responses first.

**Weaknesses:**

A major contribution is human judgment which has some issues. First, the number of subjects is small, Second, details of how experiments are conducted and information about them is missing.


Writing can be improved.
Typos here and there:
One of the most popular task —> tasks  [page 1]
We found that evaluation results from the published papers from are generally [page 3]
These methods rely on the statistics on an InceptionNet pre-trained on the ImageNet dataset. [page 4]
A limitation in this work is the reliance on human raters, which is not only expensive and time-consuming. [page 9]

Page 3 “The goal of conditional image generation is to predict an RGB image” —> I think predict is not the right word here

**Questions:**

Q: Fig 2 -> what does y axis show? No label

Q: Regarding the ImagenHub dataset: it seems like you are using data that is already been used by others. What is some researchers have already used this data to tune their models? Couldn’t you collect an independent new test set?


Q: ImagenHub Inference Library is a great job. How you ensured that the best parameter setting is chosen to generate best results for each model?


Q: What is the last row of Fig 3 showing?!


Q: In Eq. 1, why min is used? Isn’t min too stringent here? Why not mean instead!?

Q: why are there multiple errors bars for each condition? Not clear

Q:  “We assigned 3 raters for each model and computed the SC score, PQ score, and Overall human score”. 3 subjects is really not that many here and this makes the results less reliable.

Q: what is the keyword column in Table 3?

Q: How is the overall column is computed in table 4?

Q: No information about the subjects and biases etc are given.

---

> ### Author Response · Authors · 2023-11-17
> **Response to Reviewer nZYf**
>
> (w1a) A major contribution is human judgment which has some issues. First, the number of subjects is small...
>
> We studied the effect on the sample size of raters and found that 3 raters achieved high inter-rater reliability and adding more raters will not create a large impact on the mean score. The detail of this ablation study is on A6. Therefore, we stick with three raters.
>
> (w1b) ...Second, details of how experiments are conducted and information about them are missing.
>
> We detailed the experiment setup in Section 5. We provided the details of human judgment and examples in Appendix A2 and A3.
>
> (w2) Writing can be improved. Typos here and there...
>
> We now fixed the mentioned typos and improved the grammar. Thanks for pointing them out!
>
> (q1) Fig 2 -> What does y axis show? No label
>
> The y-axis in Figure 2 is the overall human evaluation score $O = \sqrt{SC\times PQ}$ in [0.0,1.0] (equation explained in Section 4.1). We have updated Figure 2 in the revised version.
>
> (q2) Regarding the ImagenHub dataset: it seems like you are using data that is already been used by others. What is some researchers have already used this data to tune their models? Couldn’t you collect an independent new test set?
>
> Most of the data sources we used (released from others) are just containing inputs without any ground truth (target). For example, all the T2I datasets are just text prompts. The DreamBench only contains some subject images and prompts, there are no pictures of what the target should look like. The same goes the multi-concept image generation and other datasets where only a set of input prompts and images. We think these are not enough to tune any model. There is still a risk that they actually hire human professionals to create outputs and tune the model on those outputs. But the cost should be pretty high.
>
> (q3) ImagenHub Inference Library is a great job. How you ensured that the best parameter setting is chosen to generate best results for each model?
>
> We used the suggested parameters contained in the official code repositories of each model's work (mentioned in Section 5). We compare our implemented model outputs against their original officially released model outputs. We ensure everything is aligned. We believe the parameters released by the original authors are already well-tuned.
>
> (q4) What is the last row of Fig 3 showing?!
>
> The last row of Figure 3 shows an OpenPose-conditioned Image Generation. In control-guided image generation tasks, the condition can be canny edges, depth maps, openpose, etc. Figure 19 shows more examples of this task.
>
> (q5) In Eq. 1, why min is used? Isn’t min too stringent here? Why not mean instead!?...
>
> * Choosing min for this purpose emphasizes the importance of meeting all criteria without exception.
> * Using mean would also increase the complexity of human user rating and introduce a misalignment across tasks, as different tasks contain a different number of conditions c_i.
>
> (q6) why are there multiple errors bars for each condition? Not clear
>
> Each error bar in a color represents a model in the specific task. The idea of Figure 4 is to show the model performance distribution in each task.
>
> (q7) “We assigned 3 raters for each model and computed the SC score, PQ score, and Overall human score”. 3 subjects is really not that many here and this makes the results less reliable...
>
> We found that 3 raters achieved high inter-rater reliability and adding more raters will not create a large impact on the mean score (refer to A6 in Appendix).
>
> (q8) what is the keyword column in Table 3?...
>
> The keyword column lists the distinctive features, techniques, or unique selling points of the model.
>
> (q9) How is the overall column is computed in table 4?...
>
> The overall column is the overall human evaluation score $O = \sqrt{SC\times PQ}$ (equation explained in Section 4.1).
>
> (q10) No information about the subjects and biases etc are given.
>
> We added the demographic information of annotators in A6.

---

> > ### Author Response · Authors · 2023-11-21
> > **Follow-up**
> >
> > Dear Reviewer nZYf,
> > I am writing to inquire if you have had the chance to review our revised manuscript and the response to your comments. We greatly value your feedback and would appreciate any further questions or comments you might have.
> > Sincerely,

---

> > > ### Author Response · Authors · 2023-11-23
> > > **2nd Follow-up**
> > >
> > > Dear Reviewer nZYf, As the author response period is ending on November 22nd, I wanted to check if you have any final comments or questions regarding our manuscript. Your feedback is greatly appreciated. Thank you,

---

### Author Response · Authors · 2023-11-17
**Changelog**

We thank all the reviewers for their thoughtful comments. We have added many experiments and generation results to the revision based on these feedbacks. Here is the change log:

- As suggested by Reviewer nZYf, we improved the grammar of the paper.
- As suggested by Reviewer nZYf, we updated Figure 2 with the y-axis label.
- As suggested by Reviewer nZYf, we added Appendix A6 to show the human evaluation procedure and the sample size consideration of human raters.
- As requested by Reviewer gWLk, we added FID and KID score in Table 7.
- As requested by Reviewer JKzK, we added MidJourney or DALLE-3 results in Datasheet B1 (Table 10).
- As pointed by Reviewer teCT, we added Appendix A7, A8, A9 to demonstrate how a third party and researchers can benefit from our work and easily extend it.

---

### Meta-Review · Area_Chair_gYkS · 2023-12-08

**Metareview:**

This paper proposes a novel benchmark suite for conditional image generation, including new evaluation datasets (composed of existing public datasets), human evluation scores and unified inference pipeline. The author rebuttal seems to address most of the points raised in the reviews. All reviewers have apositive stance on the work, although one reviewer rated the paper as slightly below the threshold, but they did not respond to the author rebuttal.

**Justification For Why Not Higher Score:**

The paper presents a useful evaluation of generative image models.
The lasting impact of the paper is unclear, as it relies on human raters, and it's not clear future works will benefit from its findings.

**Justification For Why Not Lower Score:**

All reviewers express they have a positive stance on the paper.
The only reviewer rating it as marginally below the bar, did not respond to the rebuttal.

---

### Decision · Program_Chairs · 2024-01-16

Accept (poster)